# dRTEL1 is essential for the maintenance of *Drosophila* male germline stem cells

Ying Yang[1,2,3], Ruiyan Kong[4], Feng Guang Goh[1], W. Gregory Somers[5], Gary R. Hime[5], Zhouhua Li[4], Yu Cai[1,2]*

1 Temasek Life Sciences Laboratory, National University of Singapore, Singapore, Singapore, 2 Department of Biological Sciences, National University of Singapore, Singapore, Singapore, 3 Department of Pathology, Peking University Health Science Center, Beijing, China, 4 College of Life Sciences, Capital Normal University, Beijing, China, 5 Department of Anatomy and Physiology, The University of Melbourne, Melbourne, Australia

* caiyu@tll.org.sg

**Data Availability Statement:** The RNA-seq generated during this study are available at Mendeley Data (https://data.mendeley.com/datasets/9x27fk49nm/1, https://data.mendeley.com/datasets/8zgxnk8fg8/1, https://data.mendeley.

## Abstract

Stem cells have the potential to maintain undifferentiated state and differentiate into specialized cell types. Despite numerous progress has been achieved in understanding stem cell self-renewal and differentiation, many fundamental questions remain unanswered. In this study, we identify dRTEL1, the *Drosophila* homolog of Regulator of Telomere Elongation Helicase 1, as a novel regulator of male germline stem cells (GSCs). Our genome-wide transcriptome analysis and ChIP-Seq results suggest that dRTEL1 affects a set of candidate genes required for GSC maintenance, likely independent of its role in DNA repair. Furthermore, dRTEL1 prevents DNA damage-induced checkpoint activation in GSCs. Finally, dRTEL1 functions to sustain Stat92E protein levels, the key player in GSC maintenance. Together, our findings reveal an intrinsic role of the DNA helicase dRTEL1 in maintaining male GSC and provide insight into the function of dRTEL1.

## Author summary

Adult tissue homeostasis is maintained by its resident tissue-specific stem cells. The maintenance of stem cells depends on niche activity (the so-called tissue-specific microenvironment) as well as stem cell intrinsic factors. In a genetic screen, we identify dTREL1 as an intrinsic factor required for the maintenance of *Drosophila* male germline stem cells (GSCs). Loss of dRTEL1 activity leads to premature loss of male GSCs. Our study shows that dRTEL1 likely functions through multiple downstream genes to promote GSC maintenance. Further results indicate that dRTEL1 also prevents activation of the DNA damage response pathway in GSCs for its maintenance. These factors likely converge on the regulation of Stat92E, a well-studied intrinsic factor for GSC maintenance.

com/datasets/bs72w8ynz5/1). The CHIP-seq Data generated during this study are available at Mendeley Data (https://data.mendeley.com/datasets/vyzccs8tsh/1). All other relevant data are within the manuscript and its Supporting Information files.

**Funding:** This work was supported by Temasek Life Sciences Laboratory and Singapore Millennium Foundation to Y.C., China Postdoctoral Science Foundation: 2020M670063, 2020T130030 and National Natural Science Foundation of China: 82001200 to Y.Y., and National Natural Science Foundation of China: 31972893,92054109 and Municipal Natural Science Foundation of Beijing: KZ201910028040 to Z.L. The funders had no role in study design, data collection and analysis, decision to publish, or preparation of the manuscript.

**Competing interests:** The authors have declared that no competing interests exist.

## Introduction

The balance between stem cell self-renewal and differentiation is essential for normal developmental processes as well as tissue homeostasis [1–5]. Dysregulation of stem cell self-renewal may result in stem cell over-proliferation (leading to tumor formation) or stem cell loss (resulting in premature ageing) [6–9]. Therefore, stem cell activity must be tightly controlled to maintain tissue homeostasis.

The *Drosophila* testis provides a powerful model system for the study of stem cell biology (Fig 1A) [10–12]. At the apex of the testis, 6 to 9 germline stem cells (GSCs) adhere to a group of post-mitotic stromal hub cells and each GSC is encapsulated by two somatic cyst stem cells (CySCs). The hub cells and CySCs serve as the niche for the residing GSCs and control their self-renewal and differentiation [10,12–14]. GSC divides asymmetrically to generate one GSC daughter cell, which remains in the niche (self-renewal), and one gonialblast (GB) daughter, which is located outside the niche and initiates differentiation [15]. GB undergoes 4 rounds of synchronous transit-amplifying divisions, giving rise to 16 interconnected spermatogonial cells, which further undergo two rounds of meiotic divisions to produce 64 haploid spermatids that consequently give rise to mature sperms [16].

The niche plays essential roles in maintaining GSCs by providing both physical support and self-renewing molecules, among which is Unpaired (Upd), the ligand of the Janus kinase-signal transducer and activator of transcription (JAK/STAT) signaling pathway [10,12]. Upd, produced by hub cells, binds to the receptor, Dome, on the surface of adjoining GSCs and CySCs, which in turn phosphorylates and activates Stat92E. The activated Stat92E translocates

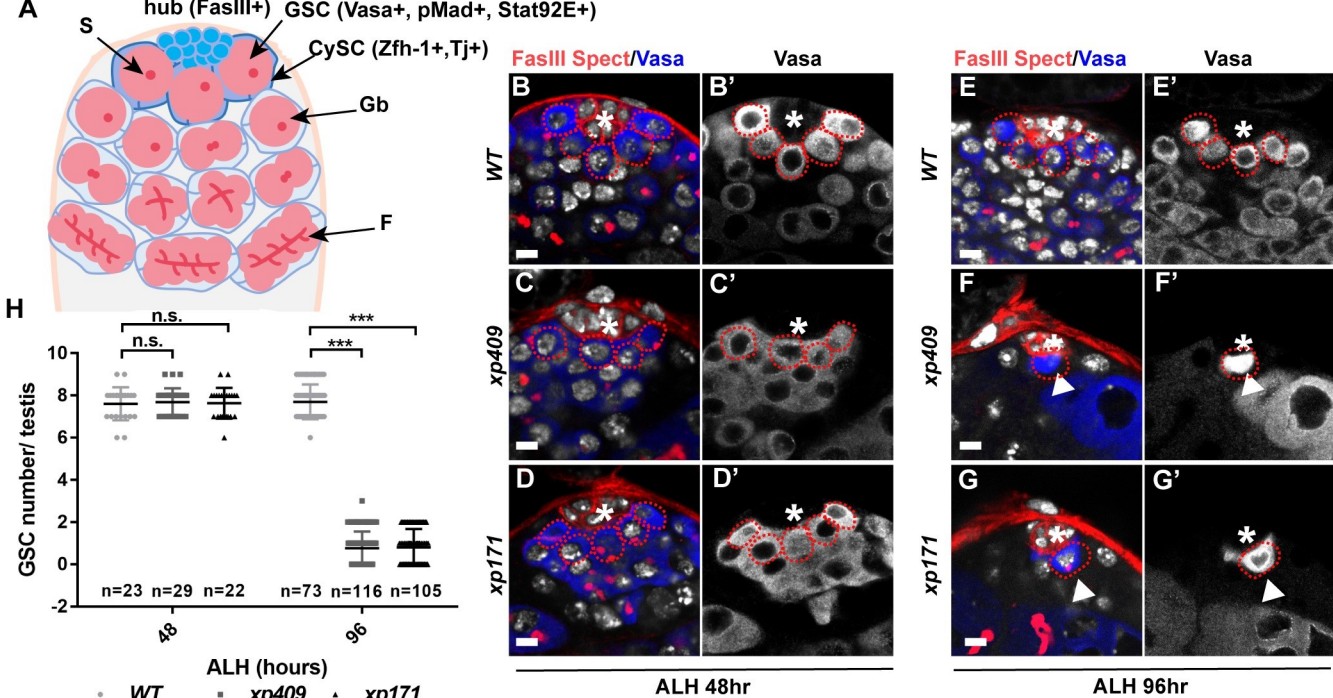

**Fig 1. dRTEL1 plays a crucial role in GSC maintenance in *Drosophila* larval gonad.** (A) A schematic diagram showing the anterior tip of *Drosophila* testis. (B-G) Representative confocal images of GSCs in *Drosophila* larval testis. ToPro-3 in white. Wild-type (control) testes maintain similar number of GSCs at 48 hr ALH (B,B') and 96 hr ALH (E,E'). *xp409* (C,C',F and F') and *xp171* (D,D',G and G') exhibit GSC loss phenotype at 96 hr ALH, compared to 48hr ALH. (H) Quantification of the average GSC number per larval testis. Number in each bar represents the number of testes examined. Data are mean±se. n.s., not significant, *, P<0.05, **, P<0.01, ***, P<0.001. WT, wild-type; GSC, Germline stem cell; GB, gonialblast; S, spectrosome; F, fusome. The hub is indicated by asterisks. GSCs are indicated by red dotted circles. Scale bar: 5 μm.

into the nucleus, subsequently regulating the transcription of Stat92E-responsive genes [17,18]. Activation of the JAK/STAT signaling pathway in CySCs and GSCs is essential for their maintenance [13,14]. Homozygous mutant of S*tat92E* exhibits defects in maintaining both GSCs and CySCs [10,12]. In addition, the niche also provides Dpp (Decapentaplegic) and Gbb (glass bottom boat), two fly BMP ligands, which bind to the receptors to activate Dpp/ BMP signaling in GSCs by converting Mad (mothers against Dpp) to pMad (phosphorylated Mad), which subsequently enters nucleus to repress transcription of the differentiation-pro- moting factor Bam (Bag of marbles) [19–23]. Apart from these signaling pathways, DE-Cad- herin-mediated attachment of GSCs to the hub also plays a crucial role for GSC maintenance [24]. Additionally, epigenetic regulators act intrinsically and contribute to the maintenance of GSCs [25,26]. The nucleosome remodeling factor (NURF) complex plays an essential role in both GSC and CySC maintenance [27]. Despite numerous progress achieved, many funda- mental questions remain unsolved.

In search for novel intrinsic regulators for male GSC maintenance and differentiation, we carried out an EMS mutagenesis screen and identified *CG4078* as a potential candidate. *CG4078* is the *Drosophila* homolog of vertebrate *Regulator of Telomere Elongation Helicase 1* (*RTEL1*) and is thus named as *dRTEL1*. *RTEL1* encodes a conserved DNA helicase-like protein which is initially identified as an essential player in regulating telomere length and maintaining the genomic stability in vertebrates [28]. Murine embryonic stem cells defective in RTEL1 function exhibit telomere loss and compromised chromosomal integrity upon induced differ- entiation [28]. *C. elegans RTEL1* functions in eliminating inappropriate homologous recombi- nation events during DNA repair [29]. *RTEL1* also accounts for the suppression of excess meiotic crossovers (CO) in *C. elegans* [30]. Moreover, mutation of *RTEL1* is often implicated in severe congenital dyskeratosis, a telomere-mediated disease [31,32]. Subsequent studies fur- ther demonstrate a key role of RTEL1 in DNA repair and genome replication and provide a mechanistic insight for this role by showing that RTEL1 suppresses the formation of G-quad- ruplex/R-loops to coordinate transcription and DNA replication [33–35].

Here, we report that dRTEL1 is a novel regulator of *Drosophila* male GSCs. Our results show that *dRTEL1* acts cell-autonomously in male germline to maintain GSCs. Transcriptome and ChIP-Seq data suggest that dRTEL1 affects a set of target genes essential for GSC mainte- nance. Further investigation reveals that dRTEL1 also functions through the DNA-damage response (DDR) pathway to maintain GSCs. Together, our data show that dRTEL1 acts through multiple downstream targets to maintain GSCs, thus revealing a novel role of the DNA helicase dRTEL1 in stem cells. Interestingly, additional data show that dRTEL1 likely promotes female GSC maintenance via different downstream targets.

## Results

### dRTEL1 plays a crucial role during larval germline development

To uncover novel regulators for male GSC maintenance and differentiation, we conducted an EMS mutagenesis screen and focused on the X-chromosome (S1A Fig). To this end, we estab- lished 2,000 pupal lethal lines and identified 21 mutants showing defective larval testis devel- opment [36]. In this study, we present our analyses on one complementation group exhibiting male GSC loss phenotype (Fig 1). This complementation group includes *xp157*, *xp171*, and *xp409*. Genetic mapping and candidate approach identified one annotated gene (*CG4078*) that was mutated in all three mutants (S1B Fig). *CG4078* encodes a RAD3/XPD-like DNA helicase which shares 66.3% similarity to the human Regulator of Telomere Elongation Helicase 1 (RTEL1) protein. Thus, *CG4078*, the *Drosophila* homolog of vertebrate *RTEL1*, is named as *dRTEL1*. The *xp157* allele bears a splicing donor mutation in the intron between the first two

coding exons (G-A mutation), which possibly leads to alternative splicing followed by a premature stop codon at base 1125. The *xp171* allele carries a premature stop codon (C-T mutation) disrupting base 2065 of the coding sequence (CDS), which locates in the conserved Helicase_C_2 domain. The *xp409* allele contains an A to T conversion at nucleotide base 547 of CDS, leading to a premature stop codon in the conserved N-terminal DEAD2 domain (S1B and S1C Fig). Confirming *CG4078/dRTEL1* is the candidate gene responsible for the observed germline phenotype in these mutants, a *dRTEL1-GFP* transgene on 2nd chromosome carrying a fosmid with GFP fused to C-terminus of dRTEL1 protein under its endogenous regulatory elements could fully rescue lethality and testis phenotype in all three EMS mutants (Figs 1B–1H and S1D–S1H) [37].

A typical *Drosophila* larval testis contains 6 to 9 GSCs adjacent to the hub located at the apical tip of the testis [38]. GSCs are identified as those pMad-positive, spectrosome-containing (a unique germline-specific organelle enriched with membrane skeletal proteins such as α-Spectrin) and Vasa-expressing germ cells in direct contact with the hub and CySCs (Fig 1A). The hub is comprised of a group of FasIII-positive post-mitotic cells, and CySCs are identified as the hub-contacting somatic cells which express the transcription factor Zn finger homeodomain 1 (Zfh1). In control, each testis contained 7.6±0.7 GSCs (n = 23) and 7.7±0.8 GSCs (n = 73) at 48 hr and 96 hr after larval hatching (ALH), respectively (Fig 1B, 1E and 1H). While *dRTEL1* testes hosted similar numbers of GSCs (7.6±0.8 in *xp409*, n = 29, and 7.6±0.7 in *xp171*, n = 22) at 48 hr ALH (Fig 1C, 1D and 1H), about 41% of *dRTEL1* testes (n = 116, for *xp409*; n = 105, for *xp171*) contained 1–2 GSCs next to the hub at 96 hr ALH (Fig 1F–1H). The remaining *dRTEL1* testes did not contain any GSC. These observations suggest that *dRTEL1* plays a crucial role in maintaining male GSCs during larval development.

## dRTEL1 is cell-autonomously required for adult GSC maintenance

dRTEL1 could function cell-autonomously in the germline or non-cell-autonomously in somatic cyst cells to promote germline development. To address these, we examined the expression of *dRTEL1* and conducted RNA *in situ* experiment in the adult testis, which exhibited same GSC loss phenotype when dRTEL1 function was compromised specifically in the germline (Fig 2). While sense probe did not pick up any specific signal, anti-sense probe detected *dRTEL1* transcripts in both germline and somatic cells (S2A and S2B Fig). Consistently, immunofluorescence analysis of testis carrying a functional *dRTEL1-GFP* transgene revealed that dRTEL1–GFP was expressed in both somatic cells (including CySCs and cyst cells) and germ cells (including GSCs and GBs) (Fig 2A for adult testis and S1D Fig for larval testis). Similar to its vertebrate homolog, *Drosophila* dRTEL1 localized to the nucleus [39,40]. To verify whether the GFP pattern is representative of expression pattern of dRTEL1, we knocked down *dRTEL1* using *UAS-dRTEL1 dsRNA* in combination with cell type-specific Gal4 drivers. Immunostaining analysis revealed that knockdown of *dRTEL1* in somatic cells (by *tj-Gal4*) or germline cells (by *nos-Gal4*) led to loss of GFP signal in the corresponding cell type (Fig 2B and 2C), showing that the GFP signal indeed reflects dRTEL1 expression.

The ubiquitous expression pattern of dRTEL1 raises the possibility that *dRTEL1* may affect GSC maintenance cell-autonomously in germ cells or non-cell-autonomously through the surrounding somatic cells. We first addressed whether dRTEL1 acts in somatic cells non-autonomously to maintain GSCs. We noted that larval *dRTEL1* testes contained fewer Zfh1-positive somatic cells as compared to those of controls, suggesting that dRTEL1 plays a role in the somatic cyst cells (S2C–S2E Fig). To investigate whether GSC loss in *dRTEL1* testes is secondary to the loss of these somatic cells, we restored *dRTEL1* expression in these somatic cells using *UASp-dRTEL1-flag* in combination with *tj-Gal4*. While the Zfh-1-positive somatic cells

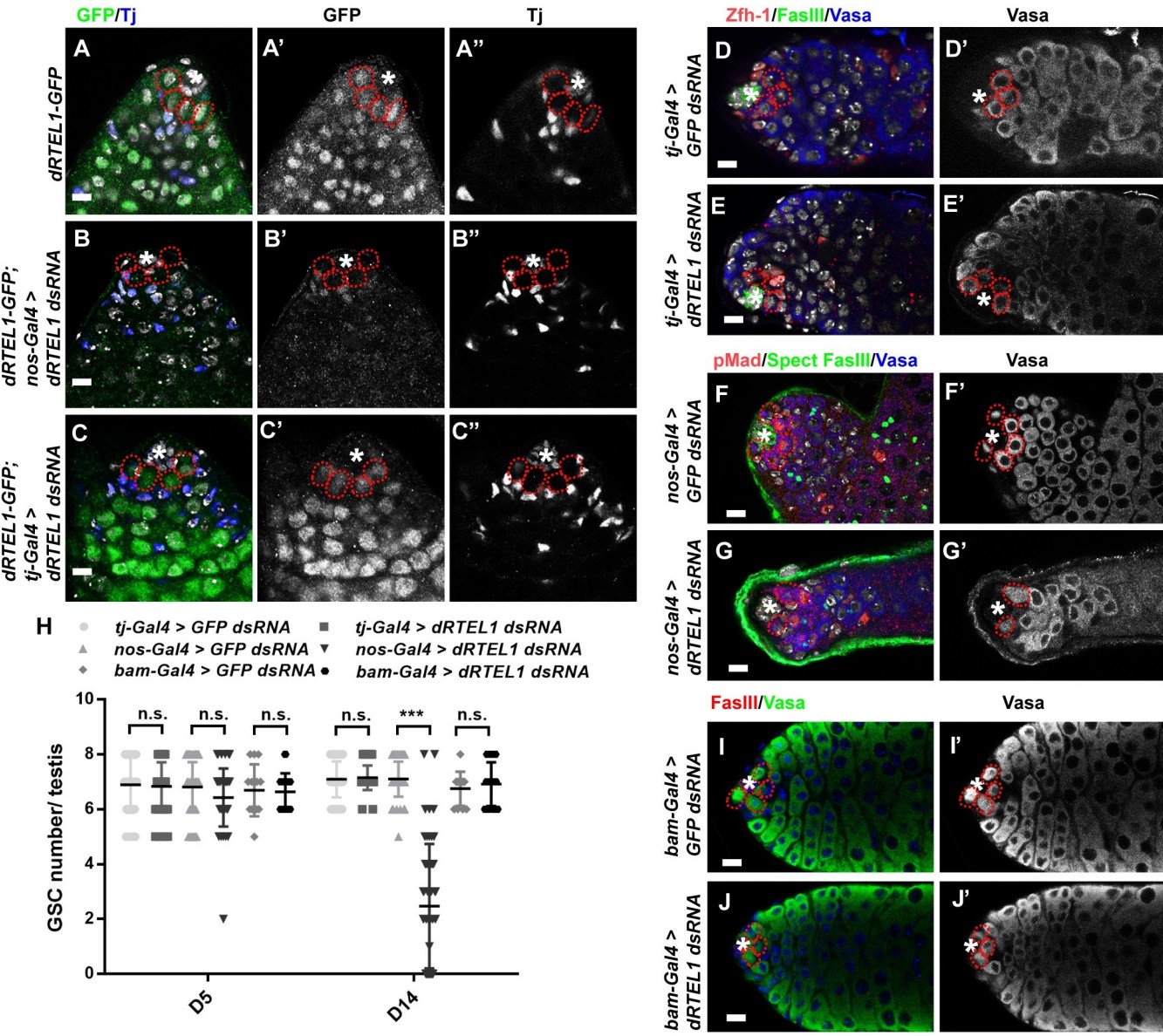

**Fig 2. dRTEL1 required cell-autonomously for male GSC maintenance.** (A-A") Representative image of day 7 (D7) *dRTEL1-GFP* adult testis showing that GFP expression in cyst cells (Tj-positive cells) and germ cells. (B-B") Representative image of D7 *dRTEL1-GFP;nos-Gal4 > dRTEL1 dsRNA* adult testis showing GFP expression in Tj-positive cyst cells but not in germ cells. (C-C") Representative image of D7 *dRTEL1-GFP;tj-Gal4 > dRTEL1 dsRNA* adult testis showing GFP expression in germ cells but not Tj-positive somatic cells. (D,D′) Representative image of D14 *tj-Gal4 > GFP dsRNA* adult testis. (E,E′) Representative image of D14 *tj-Gal4 > dRTEL1 dsRNA* adult testis. (F,F′) Representative image of D14 *nos-Gal4 > GFP dsRNA* adult testis. (G,G') Representative image of D14 *nos-Gal4 > dRTEL1 dsRNA* adult testis exhibiting GSC loss. (H) Quantification of the GSC number per testis for various genotypes. Number in each bar represents the number of testes examined. Data are mean±s.e. n.s., not significant, *, P<0.05, **, P<0.01, ***, P<0.001. (I,I′) Representative image of D14 *bam-Gal4 > GFP dsRNA* testis. (J,J′) Representative image of D14 *bam-Gal4 > dRTEL1 dsRNA* testis. The hub is indicated by asterisks. GSCs are indicated by red dotted circle. DNA (TO-PRO-3) is in white in A-G and blue in I-J. Scale bars: 5 µm (A-C and I-J) and 10 µm (D-G).

were significantly restored, these larval testes still exhibited a GSC loss phenotype, similar to those in *dRTEL1* testes (S2E–S2J Fig). To further test this, we specifically knocked down *dRTEL1* function in somatic cells using *UAS-dRTEL1 dsRNA* in combination with *tj-Gal4*. Our results showed that both larval and adult testes did not exhibit a GSC loss phenotype (Fig 2D, 2E and 2H). These data indicate the observed GSC loss in *dRTEL1* testis is unlikely due to

the loss of function in the somatic cells. We next investigated whether dRTEL1 activity is required in germ cells for GSC maintenance. For this, we compromised dRTEL1 function specifically in the germline using the combination of *UAS-dRTEL1 dsRNA* and *nos-Gal4* driver. Similar to control testes, both day 1 (newly enclosed) and day 5 testes with germline knockdown of *dRTEL1* contained 6–9 GSCs (Fig 2H). Of note, testes with compromised *dRTEL1* activity in germ cells exhibited a GSC loss phenotype at day 14 (Fig 2F–2H), indicating that dRTEL1 acts in the germline to maintain GSCs. Furthermore, testes with compromised *dRTEL1* function in differentiating spermatogonia using *UAS-dRTEL1 dsRNA* driven by *bam-Gal4* did not exhibit a GSC loss (Fig 2H–2J). Together, these data suggest that dRTEL1 functions intrinsically in male GSCs for their maintenance.

## dRTEL1 likely prevents GSC premature differentiation

The GSC loss in *dRTEL1* testis could be a consequence of cell death or premature differentiation. To distinguish these possibilities, we addressed whether *dRTEL1* GSCs underwent apoptosis using two well-established apoptotic makers, anti-cleaved Caspase-3 antibody and TUNEL assay. Anti-cleaved Caspase-3 antibody detects the active form of Caspase-3, whilst TUNEL assay detects apoptotic DNA fragments. Similar to controls, *dRTEL1* testes did not exhibit elevated signals for both apoptotic markers (Fig 3A–3F). Furthermore, the GSC loss in *dRTEL1* testes was not prevented by germline expression of Baculovirus protein p35, which effectively inhibits apoptosis in fly tissues when ectopically expressed [41–43] (Fig 3G–3J). Therefore, the GSC loss in *dRTEL1* testes is unlikely through an event of apoptosis. We next checked whether necrosis, an unprogrammed death of cells and living tissues, could be responsible for the GSC loss in *dRTEL1* mutant using propidium iodide (PI) staining, which can enter necrotic cells but is excluded from apoptotic cells [44]. However, none of the *xp409* and *xp171* GSCs examined was positive for PI (Fig 3K–3M), indicating that *dRTEL1* GSC loss is not a result of necrosis. Taken together, these data suggest that cell death is unlikely the cause of GSC loss in *dRTEL1* testis.

We then investigated whether dRTEL1 maintains GSCs by preventing its precocious differentiation. In WT, both GSCs and GBs contained a spherical-shaped fusome (also called spectrosome) and differentiating spermatogonial cysts processed a branched fusome [45,46]. While control larval testes harbored spectrosome-containing GSCs adhering to the hub cells (Fig 3N), in 25% of *dRTEL1* larval testes (n = 140 for *xp409*), branched fusomes were observed in germline cells next to the hub cells, indicative of differentiating spermatogonial cysts (Fig 3O). In WT, Stat92E is enriched in GSCs and GBs, but rarely detected in differentiating spermatogonia. While control larval testes showed strong Stat92E signals in GSCs and some GBs (Fig 3P), Stat92E was clearly reduced in the germ cells next to the hub in *dRTEL1* larval testes (Fig 3Q). Similarly, branched fusomes and decreased Stat92E expression were also observed in adult testes with compromised dRTEL1 activity specifically in the germline using *UAS-dRTEL1 dsRNA* in combination with *nos-Gal4* driver (Fig 3R–3U). Collectively, these results suggest that dRTEL1 functions in the germline to prevent precocious GSC differentiation.

## dRTEL1 affects the expression of genes required for GSC maintenance

To further understand how dRTEL1 regulates GSC maintenance at the molecular level, we performed transcriptome analyses for control larval testes, *dRTEL1* larval testes, and *dRTEL1* larval testes rescued with *dRTEL1-GFP* to identify genes affected by dRTEL1 activity. The samples were collected from early L3 testes (60–72 hr) when *dRTEL1* testes still contained many GSCs and GBs to minimize the difference between control and *dRTEL1* testes (S3A and S3B Fig). Of the 12,228 annotated genes detected in these samples, about 15% of the genes (1,817)

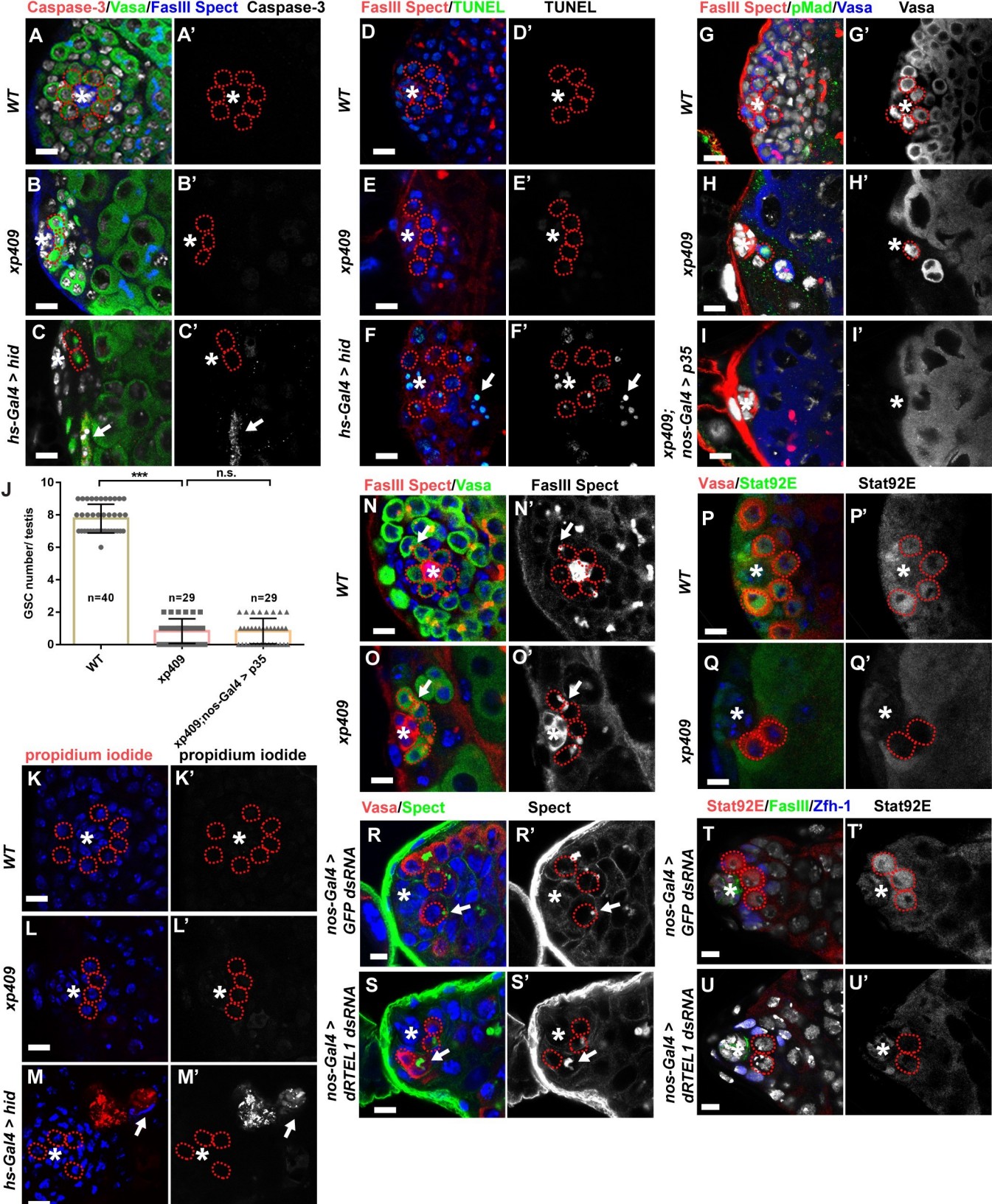

**Fig 3. dRTEL1 maintains GSCs by preventing precocious differentiation.** (A-C') Representative images of 72 hr ALH WT or *xp409* or *hs>hid* larval testis stained with anti-cleaved Caspase-3. N = 50 for each phenotype. Arrow in C,C' indicates one Caspase-3-positive germline cyst. (D-F') Representative images of 72 hr ALH WT or *xp409* or *hs>hid* larval testis with TUNEL staining. N = 50 for each phenotype. Arrow in F,F' indicates TUNEL-positive germline cyst. (G-I') Representative images of 96 hr ALH WT (G,G'), *xp409* (H,H') or *xp409;nos-Gal4 > p35*(I,I') larval testis. (J) Quantification of average GSC number per testis in these genetic backgrounds. Number in each bar represents the number of testes examined. Data are mean±s.e. n.s., not significant, *, P<0.05, **, P<0.01, ***, P<0.001. (K-M) Representative image of 72 hr ALH WT (K,K') or *xp409* (L,L') or *hs>hid* larval testis (M,M') stained with Propidium iodide (PI). N = 20 for each phenotype. Arrow in M,M' indicates PI-positive cyst. (N,N') A WT larval testis at 72 hr ALH harboring round spectrosome (arrow) in GSCs. (O,O') A *xp409* larval testis at 72 hr ALH containing branched fusome (arrow) in germ cells next to the hub. (P,P') A WT larval testis at 72hr ALH showing Stat92E expression in GSCs, GBs and CySCs. (Q,Q') A *xp409* larval testis at 72hr ALH showing reduced Stat92E expression in GSCs. (R,R') A D7 *nos-Gal4 > GFP dsRNA* testis showing round spectrosome-containing germ cells (arrow) next to the hub. (S,S') A D7 *nos-Gal4 > dRTEL1 dsRNA* testis showing branched fusome (arrow) in germ cells next to the hub. (T,T') A D14 *nos-Gal4 > GFP dsRNA* testis showing Stat92E expression in GSCs. (U,U') A D14 *nos-Gal4 > dRTEL1 dsRNA* testis exhibiting reduced Stat92E expression in GSCs. The hub is indicated by asterisks. GSCs are indicated by red dotted circles. DNA (TO-PRO-3) is in white in A-C, G-I and T-U and blue in D-F, K-S. Scale bars: 5 μm.

were differentially expressed between *dRTEL1* and control testes, and their expression levels were restored to the control level in *dRTEL1* testes rescued with *dRTEL1-GFP* (Fig 4A), indicating that these genes are downstream targets of dRTEL1. Among these candidate genes, 831 (45.7%) were up-regulated (Fig 4B and S1 Table) and 986 (54.3%) were down-regulated in *dRTEL1* testes (Fig 4B and S2 Table). Gene ontology (GO) term analysis of these RNA-Seq data revealed that genes belonging to biological processes such as biosynthetic process, cell differentiation, response to stress, and reproduction were differentially expressed in *dRTEL1* testes (S3C and S3D Fig). To verify these RNA-Seq data, we randomly selected 21 genes and conducted quantitative reverse transcriptase (qRT)-PCR from these RNA sequencing samples. For all genes examined, RNA-Seq and qRT-PCR results showed the same trend of altered expression although the exact fold changes in transcription levels exhibited some variations (S3E Fig). Thus, the RNA-Seq data are of high quality and accurately reflect the transcriptional differences between *dRTEL1* and control testes. Our results are consistent with one recent publication showing that loss of *RTEL1* in mouse embryonic fibroblasts (MEFs) resulted in extensive transcriptional alteration [35].

While no ectopic expression screen has been carried out, several large-scale germline knockdown screens have been conducted in fly germline [47,48]. Thus, we proceeded to compare those downregulated genes with the candidates implicated in GSC maintenance identified in those knockdown screens and did not pursue those upregulated genes in *dRTEL1* mutant. Among those 986 down-regulated genes, 136 genes were previously identified required for ovary development [47] (Fig 4C and S3A Table). 38 genes were identified to be essential for male GSC maintenance [48] (Fig 4D and S3B Table). 16 genes were commonly identified in these two screens, suggesting that they are candidate genes required for GSC maintenance (Figs 4E and S3F and S3C Table).

## dRTEL1 maintains male GSCs independent of its role in preventing G4/R-Loops formation

RTEL1 is a paralog of Xeroderma pigmentosum group D (XPD) and a member of FeS cluster-containing Rad3/Rchl1-like DNA helicases, with XPD being the founding member [49] (S4A Fig). This subclass helicase contains XPD, RTEL1, ChlR1, and FANCJ. XPD is a subunit of the transcription factor II H (TFIIH) and plays a dual role in transcriptional initiation and nucleotide excision repair [49]. Since dRTEL1-GFP also exhibited nuclear localization (Figs 2A and S1D) and one recent study showed that human RTEL1 colocalizes with active RNA polymerase II (RNAP II) bearing CTD domain phosphorylated at Serine 2 residue, a hallmark of productive elongation during transcription [34], we wondered whether dRTEL1 associates with the chromatin and conducted chromatin immunoprecipitation experiments followed by high-throughput sequencing (ChIP-Seq) for *dRTEL1* testes rescued with *dRTEL1-GFP*. For these

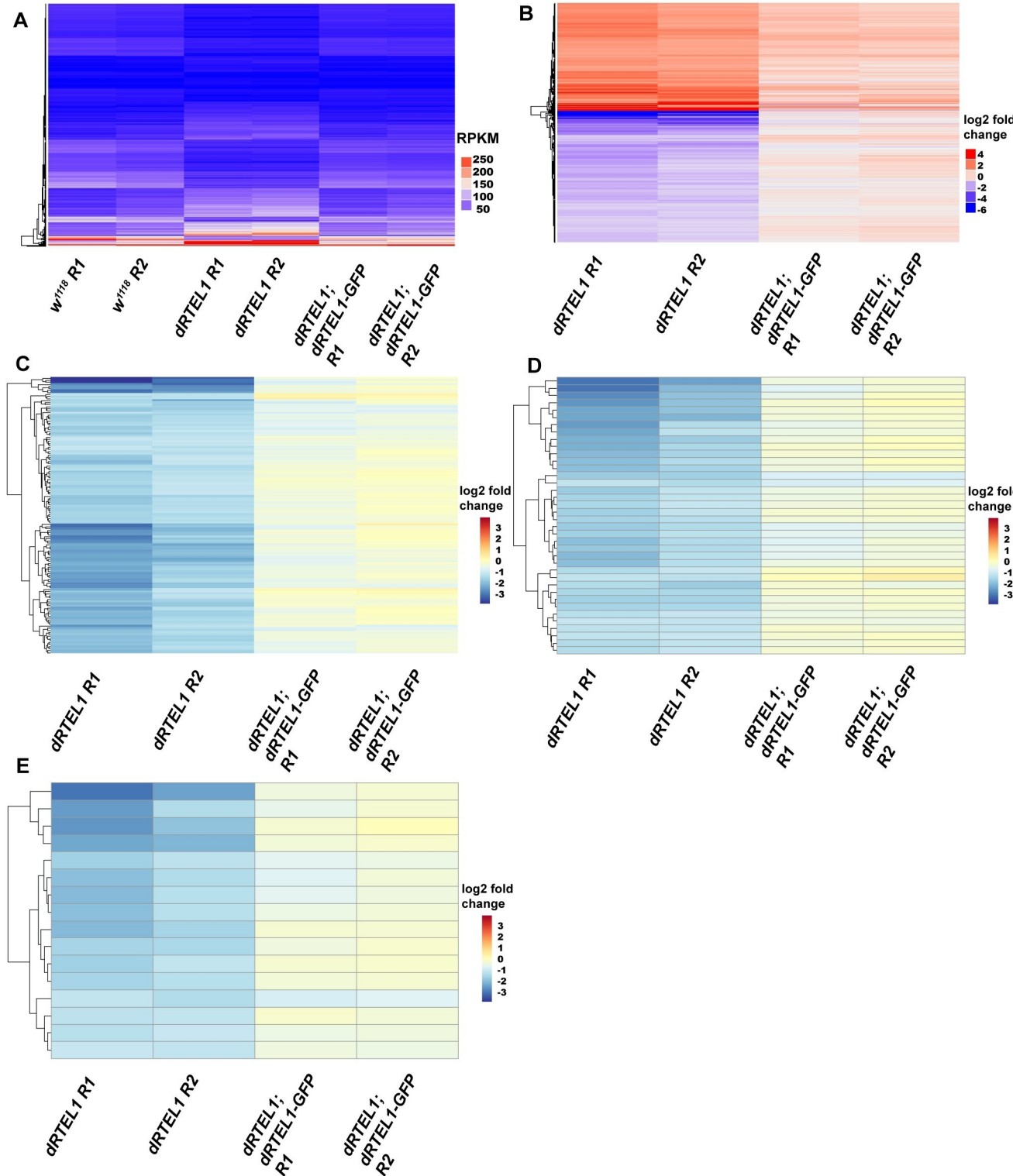

**Fig 4. dRTEL1 affects expression of genes involved in GSC self-renewal and differentiation.** (A and B) Summary of RNA-seq data. (A) The heatmap of expression level of 1,817 differentially expressed genes in *w¹¹¹⁸*, *dRTEL1* and *dRTEL1; dRTEL1-GFP* larval testes. (B) Heatmap of log2 fold change of gene expression profiles in *dRTEL1* and *dRTEL1; dRTEL1-GFP* testis. For all samples, comparisons were made to *w¹¹¹⁸* larval testes. (C) Heatmap of log2 fold change of gene expression profiles of 136 genes reported to be required for female germline development in *dRTEL1* and *dRTEL1; dRTEL1-GFP* larval testes. (D) Heatmap of log2 fold change of gene expression profiles of 38 genes identified to cause the male GSC maintenance defect in *dRTEL1* and

*dRTEL1; dRTEL1-GFP* larval testes. (E) Heatmap of log2 fold change of gene expression profiles of 16 genes identified in both RNAi screens in *dRTEL1* and *dRTEL1; dRTEL1-GFP* larval testes. R1 for Replicate 1, R2 for Replicate 2.

experiments, we used adult testes with the following two reasons. First, dRTEL1 functions in both larval and adult testes for GSC maintenance (Fig 2). Second, it's a challenging task to collect enough larval testis samples for this experiment, compared to adult testes. After normalization using GFP as a control, dRTEL1-GFP was found to be enriched at 654 genomic loci with 11.8% of the enrichment located within a 2-kb region upstream (5.8%) or downstream (6%) of gene coding region, 15.9% of the enrichment detected in coding region and 67.5% in the intronic region (Fig 5A). These data, together with the data showing that vertebrate RTEL1 colocalizes with active RNAP II, suggest that dRTEL1 might play a role in transcription. Gene ontology (GO) term analysis revealed that genes implicated in anatomical structure development, cell differentiation, and signal transduction represented the top three categories of these dRTEL1 target genes (S4B and S4C Fig).

Among those 654 putative target genes, 22 genes were down-regulated and 49 genes were up-regulated in *dRTEL1* testes from our transcriptome analysis (S4 Table). For instance, dRTEL1 was found to be enriched at the gene body region of *Nipped-A* (Fig 5B), the fly homolog of transformation/transcription domain-associated protein (TRRAP) and a subunit of histone acetyltransferase complex. *Nipped-A* transcripts were downregulated by 65% in *dRTEL1* testes ($P<0.01$) in our transcriptome analysis (Fig 5C).

To explore whether dRTEL1 might act through these downregulated genes to regulate GSC maintenance, we conducted a small-scale RNAi screen to address the function of these downregulated genes. Our results show that germline-specific knockdown of 9 genes, including *Nipped-A* (a subunit of the Tip60 chromatin-remodeling complex), *mst* (misato, a co-factor of the TCP-1 tubulin chaperone complex), *CG3527* (a putative S-adenosyl-L-methionine-dependent pseudouridine N(1)-methyltransferase), *ckn* (caskin, a cytoplasmic adaptor protein), *mld* (molting defective, a nuclear zinc finger protein), *CG34109*, *tun* (tungus, a protein with N-terminal glutamine amidohydrolase activity), *CG7504* (senataxin, a protein involves in termination of RNAP II transcription), and *CG12926* (an annotated protein with phosphatidylinositol bisphosphate binding activity), resulted in a GSC loss phenotype, reminiscent of those *dRTEL1* mutants or germline knockdown testes (Figs 5D–5F and S4D and S5 Table). We then examined the expression of these genes in testes with germline-specific knockdown of *dRTEL1*. Our results showed that the expressions of these genes were downregulated in *dRTEL1* germline knockdown testes (S4E Fig), supporting our early RNA-seq data and confirming that dRTEL1 affects the expression of these genes in the germline. Together, these data suggest that dRTEL1 might regulate GSC maintenance through promoting the expression of these candidate genes.

We further tested whether dRTEL1 could promote the expression of these candidate genes upon ectopic expression in the germline using *UASp-dRTEL1-flag* in combination with *nos-Gal4*. Interestingly, *CG34109* and *Nipped-A* were significantly upregulated in these dRTEL1-overexpressing testes (S4F Fig). Together with the report showing that RTEL1 colocalizes with active RNAP II in transcriptional elongation, these data support the notion that dRTEL1 might involve in the expression of these candidate genes.

The low overlapping between genes identified in transcriptome analysis and ChIP-seq experiments indicates that the altered expression of most genes could be an indirect consequence of reduced dRTEL1 activity, as suggested by an early report in RTEL1 knockout mouse MEFs [35]. It's further shown that in mouse MEFs, RTEL1 maintains transcription fidelity by preventing double-strand breaks (DSBs) caused by the formation of G4/R-Loops and the

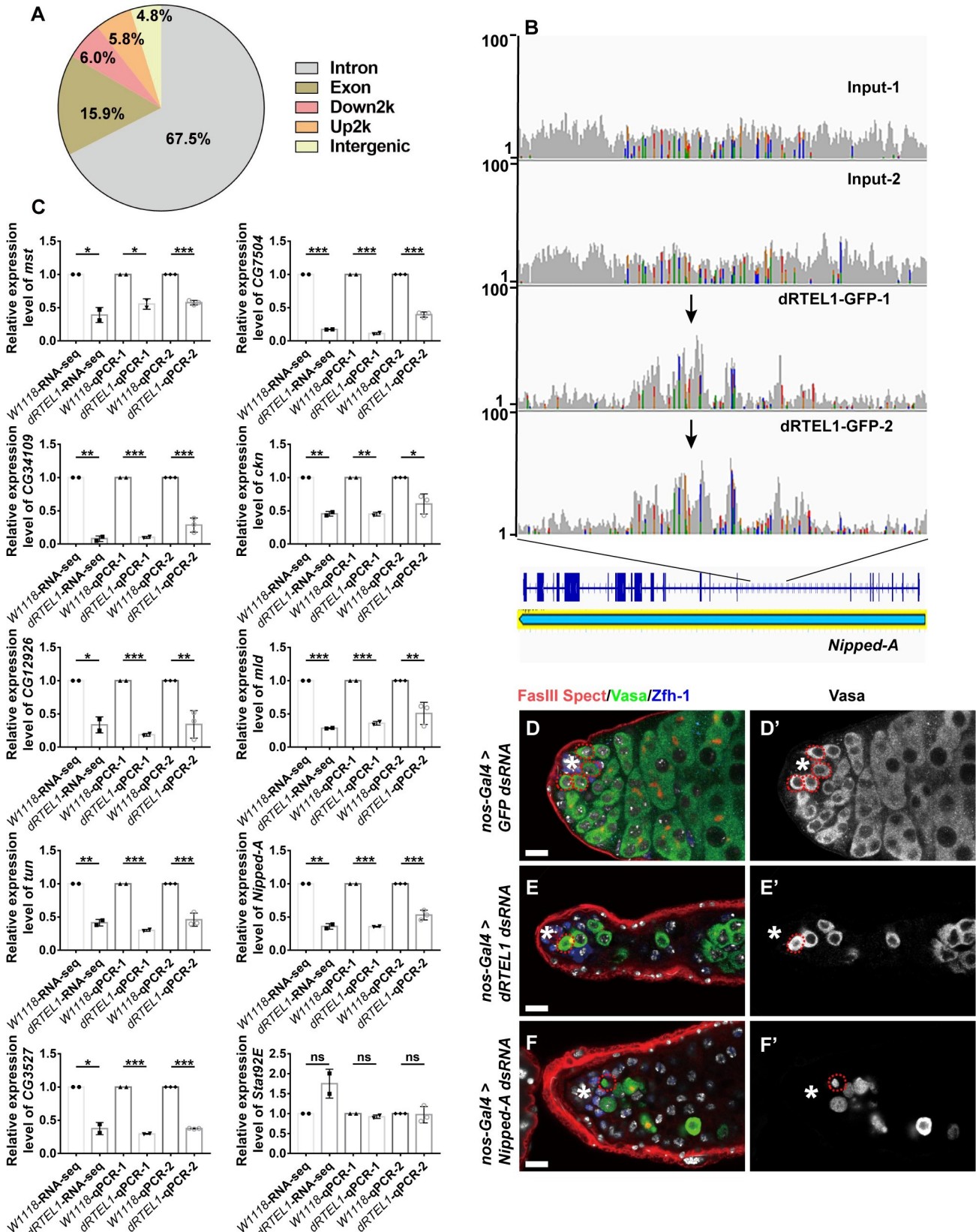

**Fig 5. dRTEL1 regulates expression of self-renewal and differentiation factors.** (A) Summary of ChIP-seq data. 67.5% of the of dRTEL1 binding sites was enriched in intron region, while 15.9%, 6%, 5.8% and 4.8% in exon, 2-kb downstream region, 2-kb upstream region and intergenic region, respectively. (B) The snapshot of genome browser results of the binding peaks of dRTEL1-GFP on *Nipped-A*. (C) Verification of RNA-Seq results by qRT-PCR for selected genes. Black bar indicates fold change obtained from RNA-seq data. qPCR1 is performed using samples for RNA-seq, qPCR2 is conducted using different batch of RNA sample prepared. (D,D′) Representative image of D14 *nos-Gal4 > GFP dsRNA* adult testis. (E,E') Representative image of D14 *nos-Gal4 > dRTEL1 dsRNA*. (F,F') A D14 *nos-Gal4 > Nipped-A dsRNA* adult testis exhibiting GSC loss. Data are mean±s.e. n.s. not significant, *, P<0.05, **, P<0.01, ***, P<0.001. The hub is indicated by asterisks. GSCs are indicated by red dotted circles. DNA is in white (TO-PRO-3). Scale bars: 10 μm.

similar transcriptional alteration was observed in MEFs treated with 10 μM TMPyP4, the G4-DNA/R-Loops stabilizer [35]. We wondered whether the aberrant expression of these candidate genes in *dRTEL1* mutant was a result of DSBs caused by G4/R-Loops formation. When dRTEL1 function was compromised in S2 cells via dsRNA-mediated knockdown, the expression of those downstream genes (except *CG3527* and *Nipped-A*) remained unchanged (S4G Fig), suggesting that dRTEL1 might affect gene expression in a context-dependent manner. Of note, S2 cells treated with various concentrations of TMPyP4 (1 μM, 10 μM, or 50 μM) did not exhibit transcriptional changes for all 9 candidate genes (S5A Fig), indicating that dRTEL1 affects the expression of these genes independently of G4/R-Loop formation in S2 cells. To gain further support, we fed *Drosophila* larvae with various concentrations of TMPyP4 throughout development upon larvae hatching and examined its effect on male GSC maintenance. To monitor DSBs induced by TMPyP4, we used two well-established DSB markers, γ-H2Av (the *Drosophila* equivalent of mammalian γ-H2AX) and p53 (the cell cycle regulator which is involved in DNA damage checkpoints in metazoan cells). As expected, TMPyP4 feeding resulted in ectopic accumulation of both DSB markers in L3 larval GSCs (Fig 6A–6D). These TMPyP4-treated L3 testes however, harbored [6.5±0.2 (n = 12) for 1 μM, 6.6±0.2 (n = 11) for 10 μM, and 6.4±0.2, n = 14 for 50 μM] GSCs, similar to that of control testes (6.7 ±0.2, n = 10), but distinct from that of *dRTEL1* testes (0.8±0.1, n = 116) (Figs 6E and 1H). Surprisingly, these larvae could successfully develop into adult. Of note, these adult testes contained similar number of GSCs as controls (6.7±0.3, n = 17 for 1 μM, 6.9±0.2, n = 17 for 10 μM, 6.3±0.2, n = 34 for 50 μM, and 6.6±0.2, n = 22 for control) and did not exhibit GSC loss, showing that DSBs induced by TMPyP4 alone is not sufficient to result in GSC loss (Fig 6F–6J). Supporting this, both TMPyP4-treated larval and adult testes did not show transcriptional changes for those candidate genes whose expressions were downregulated in testes of germline-specific knockdown of *dRTEL1* (S5B and S5C Fig). These results suggest that dRTEL1 likely maintains male GSCs independently of its presumptive role in DSBs induced by G4 formation.

## dRTEL1 maintains GSCs by preventing DNA damage-induced checkpoint activation

Ma et al., reported previously that DNA damage in *Drosophila* ovarian GSCs leads to precocious GSC loss [50]. Similarly, mouse MEFs with loss of RTEL1 exhibited accumulation of DSB markers γ-H2Av and 53BP1 [35]. Our early results also show that TMPyP4 treatment induced DSBs in fly larval and adult testes (Fig 6). As such, we further addressed whether dRTEL1 also plays a role in DNA damage repair in male germline and whether the DNA damage response (DDR) pathway has a role in GSC loss observed in *dRTEL1* testes [51].

While γ-H2Av signal is absent from control GSCs and GBs, it was strongly accumulated in *dRTEL1* GSCs and GBs (Fig 7A, 7B and 7F), indicative of elevated DNA damage. To further address the activation of the DDR pathway in *dRTEL1* mutant, we examined the activity of p53, which is activated in *Drosophila* ovarian GSCs in response to DNA damage [52,53]. The

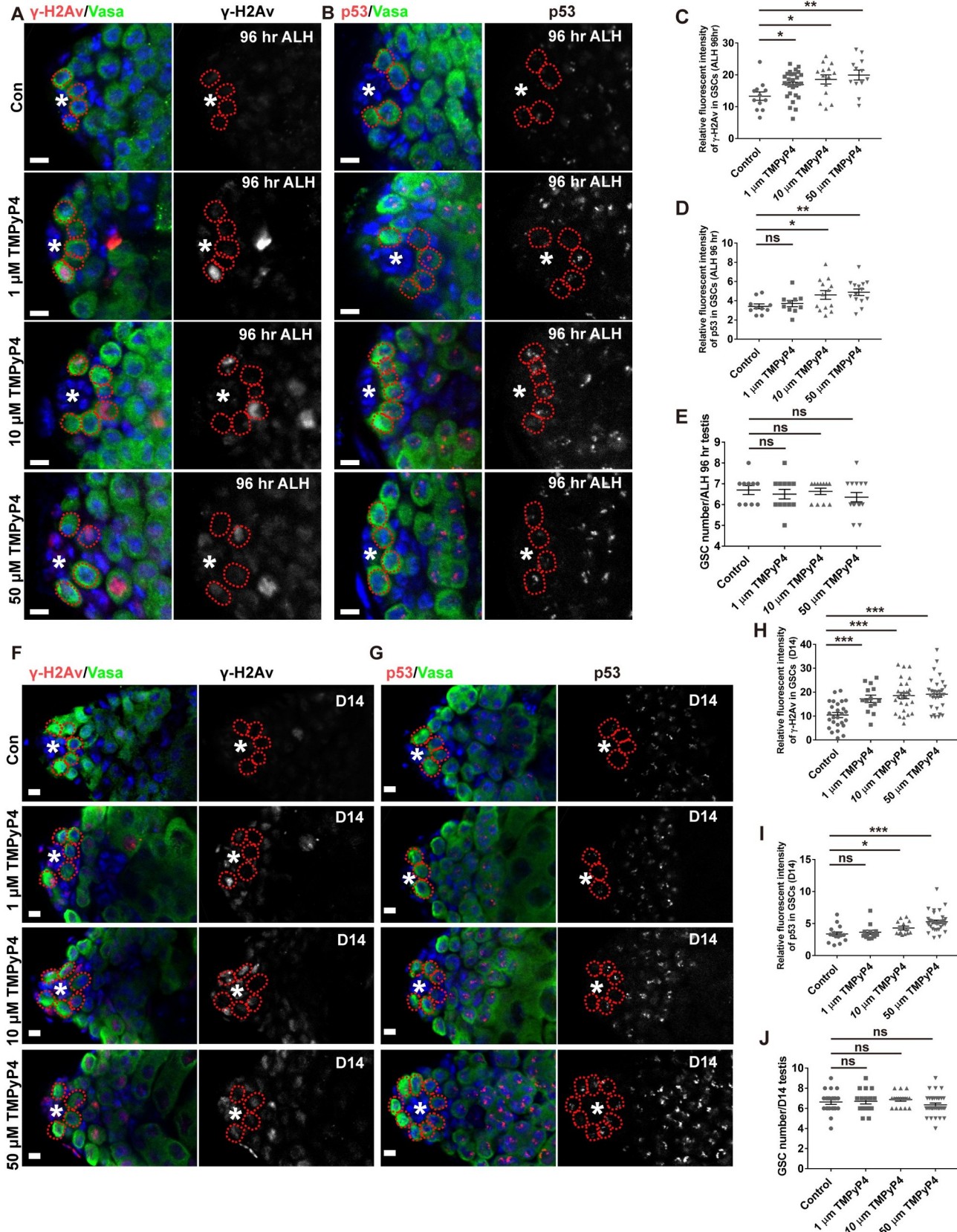

**Fig 6. dRTEL1 maintains male GSCs independent of its role in preventing G4/R-Loops formation.** (A) γ-H2Av expression in 96 hr ALH WT larval testis treated with 0, 1µM, 10µM and 50 µM TMPyP4. (B) p53 expression in 96 hr ALH WT larval testis treated with 0, 1µM, 10µM and 50 µM TMPyP4. (C) Quantification of the relative fluorescent intensity of γ-H2Av per GSC in various backgrounds at 96 hr ALH. (D) Quantification of the relative fluorescent intensity of p53 per GSC in various backgrounds at 96 hr ALH. (E) Quantification of GSC numbers per testis in various backgrounds at 96 hr ALH. (F) γ-H2Av expression in D14 WT adult testis treated with 0, 1µM, 10µM and 50 µM TMPyP4. (G) p53 expression in D14 WT adult testis treated with 0, 1µM, 10µM and 50 µM TMPyP4. (H) Quantification of the relative fluorescent intensity of γ-H2Av per GSC in various backgrounds at D14. (I) Quantification of the relative fluorescent intensity of p53 per GSC in various backgrounds at D14. (J) Quantification of GSC numbers per testis in various backgrounds at D14. Data are mean±s.e. n.s., not significant, *, P<0.05, **, P<0.01, ***, P<0.001. The hub is indicated by asterisks. GSCs are indicated by red dotted circles. DNA(TO-PRO-3) is in blue. Scale bars: 5 µm.

p53 activity was reported by *in vivo* biosensors that are constructed by placing the GFP coding sequence under the control of a *reaper* (*rpr*) enhancer, which contains a p53 consensus binding site [52,54]. To rule out technical artifacts, we deployed two different GFP biosensors, p53R-GFPnls that localizes to the nucleus and p53R-GFPcyt that exhibits cytoplasmic localization. Similar to adult testes described previously [53], GFP signals were not detected in control larval testes, although p53 protein expressed in germ cells including GSCs and GBs (Figs 7G, 7K, 7L and S6A). In *dRTEL1* larval testes, p53 protein level was upregulated and robust p53 biosensor activities were observed in GSCs and GBs for both reporters (Figs 7H, 7K, 7M and S6B). Occasionally, p53 reporter activity was observed in early spermatogonial cysts, possibly reflecting perdurance of GFP protein. These results indicate that p53 activity is elevated in *dRTEL1* mutant germ cells.

It was reported that DNA damage in female GSCs resulted in precocious differentiation in a CHK2-dependent manner [50]. Although our early results showed that *Drosophila* larval and adult testes with DSBs induced by TMPyP4 did not exhibit a GSC loss phenotype, the elevated γ-H2Av signal and p53 activation in *dRTEL1* male GSCs raise the possibility that DNA damage response might contribute to the observed GSC loss. To address this, we investigated the genetic interaction between *dRTEL1* and the DDR pathway components.

In eukaryotic cells, DNA damage leads to the activation of the conserved kinases ATM (Ataxia telangiectasia mutated, *tefu* in fly) and ATR (ATM-RAD3 related, *mei-41* in fly), which in turn act through downstream effector kinases CHK2 (*loki* in fly) and CHK1 (*grp* in fly), respectively, to initiate the DDR signaling pathway [55–57]. Similar to controls, day 1 adult testes with germline knockdown of *dRTEL1* contained 6–9 GSCs. Whilst day 14 testes with *dRTEL1* knockdown showed GSC loss (Figs 7N–7Q and S6C–6E). Day 1 testes with either germline single knockdown of *tefu* or double knockdown of *dRTEL1* and *tefu* did not contain any GSCs (S6F and S6G Fig), suggesting that ATM is epistatic to dRTEL1. Like what we observed in control testes, testes with germline knockdown of *mei-41* contained 6–9 GSCs (Fig 7R), which were similar to testes with germline double knockdown of *mei-41* and *dRTEL1* (6–9 GSCs at day 1). However, testes with double knockdown of *mei-41* and *dRTEL1* exhibited a GSC loss at day 14 (Fig 7S), which is similar to testes of *dRTEL1* single knockdown (Fig 7Q) and hence suggests that *mei-41* is dispensable for the GSC loss in *dRTEL1* mutant testes.

We next investigated the role of two effector kinases, Grp and Loki, in dRTEL1-mediated GSC loss. Similar to the controls, testes with germline knockdown of *grp* contained 6–9 GSCs (Fig 7T), which were Stat92E-positive (Fig 8M), γ-H2Av-negative (Fig 7E) and exhibited normal p53 level (Fig 7X). Interestingly, germline knockdown of *grp* partly prevented GSC loss in *dRTEL1* knockdown testes (Fig 7U), indicating that *grp* is responsible for the GSC loss in the absence of dRTEL1 activity. Loki (lok), the *Drosophila* homolog of the CHK2 kinase [58], plays a central role in p53 activation following DNA damage [59–62]. The *lok^P6* is a mutant deleting the translational start codon and this loss of function of *lok* suppresses the checkpoint activation-induced dorsal-ventral defect during oogenesis in "spindle-class" mutants caused by DNA damage and also suppresses DNA damaged-induced GSC loss in female germline

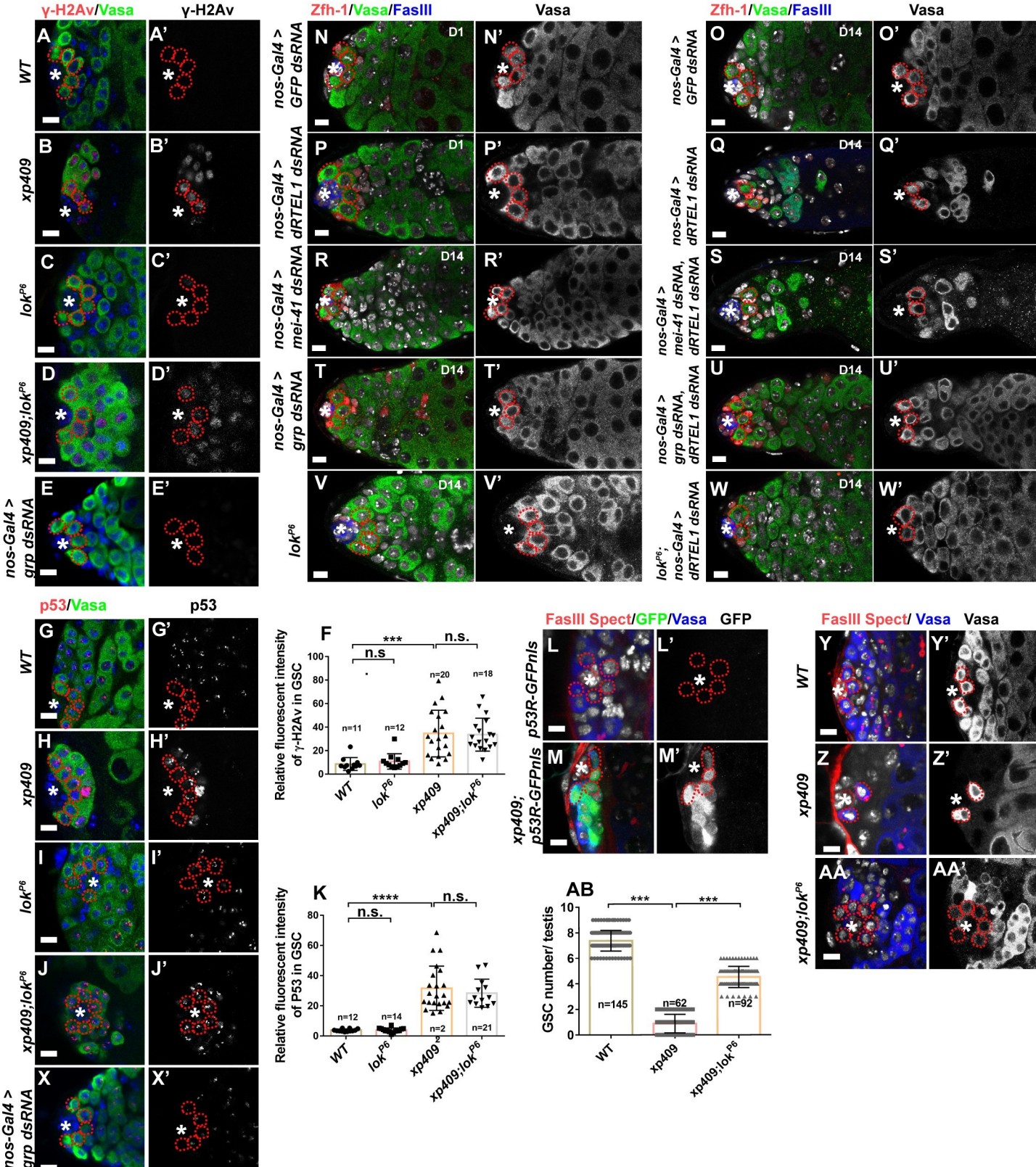

**Fig 7. dRTEL1 maintains GSCs by preventing DNA damage-induced checkpoint activation.** (A,A') No γ-H2Av expression detected in WT larval testis at 72hr ALH. (B,B') A *xp409* larval testis at 72hr ALH showing γ-H2Av accumulation in GSCs. (C,C') A *lok^P6* larval testis at 72 hr ALH showing no γ-H2Av in GSCs. (D,D') A *xp409;*

$lok^{P6}$ larval testis at 72 hr ALH showing γ-H2Av accumulation in GSCs. (E,E') A D14 *nos-Gal4 > grp dsRNA* adult testis showing no γ-H2Av in GSCs. (F) Quantification of the relative fluorescent intensity of γ-H2Av per GSC in various backgrounds. Number in each bar represents the number of testes examined. (G,G') A p53 staining in WT larval testis at 72hr ALH. (H,H') A *xp409* larval testis at 72hr ALH exhibiting elevated p53 staining in GSCs. (I,I') p53 staining in $lok^{P6}$ larval testis at 72hr ALH. (J,J') A *xp409; $lok^{P6}$* larval testis at 72hr ALH exhibiting elevated p53 staining in GSCs. (K) Quantification of the relative fluorescent intensity of p53 per GSC in various backgrounds. Number in each bar represents the number of testes examined. (L,L') A *p53R-GFPnls* larval testis at 72hr ALH showing no GFP expression in GSCs. (M,M') A *xp409; p53R-GFPnls* larval testis at 72hr ALH showing GFP expression in GSCs. (N-Q') D1 *nos-Gal4 > GFP dsRNA* (N,N') and D14 *nos-Gal4 > GFP dsRNA* (O,O') adult testis contain 6–9 GSCs. (P,P') D1 *nos-Gal4 > dRTEL1 dsRNA* adult testis contains 6–9 GSCs but (Q,Q') D14 *nos-Gal4 > dRTEL1 dsRNA* adult testis exhibiting a GSC loss. (R,R') A D14 *nos-Gal4 > mei-41 dsRNA* adult testis containing 6–9 GSCs. (S,S') A D14 *nos-Gal4 > mei-41 dsRNA, dRTEL1 dsRNA* adult testis exhibiting a GSC loss. (T,T') A D14 *nos-Gal4 > grp dsRNA* adult testis containing 6–9 GSCs. (U,U') A D14 *nos-Gal4 > grp dsRNA, dRTEL1 dsRNA* adult testis containing 5–8 GSCs. (V,V') A D14 $lok^{P6}$ adult testis. (W,W') A D14 $lok^{P6}$; *nos-Gal4 > dRTEL1 dsRNA* testis contains 6–9 GSCs. (X,X') p53 staining in *nos-Gal4 > grp dsRNA* adult testis at D14. (Y, Y') A representative image of *WT* larval testis at 96hr ALH. (Z,Z') A representative image of *xp409* larval testis at 96hr ALH containing few GSCs. (AA,AA') A representative image of *xp409;$lok^{P6}$* larval testis at 96hr ALH. (AB) Quantification of the GSC number per larval testis in various backgrounds. Number in each bar represents the number of testes examined. Data are mean±s.e. n.s., not significant, *, P<0.05, **, P<0.01, ***, P<0.001. The hub is indicated by asterisks. GSCs are indicated by red dotted circles. DNA (TO-PRO-3) is in white in L-Z and blue in A-E and G-J. Scale bars: 5 μm.

[50,63–65]. Similar to controls, *lok* testes sustained 6–9 GSCs (Fig 7V), which were negative for γ-H2Av (Fig 7C). These results suggest that *lok* is dispensable for male GSC maintenance. Interestingly, the GSC loss phenotype in *dRTEL1* testes was partially but significantly suppressed in *lok* and *dRTEL1* double mutant, although the testes were still morphologically distinguishable from controls (Fig 7Y–7AB). Of note, both GSCs and GBs of *dRTEL1* and *lok* double mutant testes still exhibited elevated γ-H2Av signal (Fig 7D), consistent with the notion that *lok* acts downstream of DSBs. Similarly, *lok* homozygous mutant significantly suppressed the GSC loss observed in *nos-Gal4>UAS-dRTEL1 dsRNA* adult testes (Figs 7W and S6I). Collectively, these results show that these two downstream effector kinases of the DDR pathway are responsible for GSC loss in *dRTEL1* mutant testes.

Our early data identified several downstream candidate genes of dRTEL1, we further investigated whether those genes might function through the DDR pathway for GSC maintenance. To this, we examined DSB marker γ-H2Av in the germline with compromised functions of some of these candidate genes. Of note, compromising these candidates (including *Nipped-A*, *CG3527*, and *mst*) in the germline did not result in elevated γ-H2Av levels (S6J––6M Fig). Although testes with germline-specific knockdown of *CG7504* exhibited ectopic γ-H2Av signal in differentiating germ cells, no elevated γ-H2Av signal was detected in GSCs (S6N Fig). These data suggest that these candidate genes do not work upstream of DSBs for the maintenance of male GSCs. In addition, the expression of these genes did not change in TMPyP4-treated S2 cells, larval or adult testis (S5 Fig), indicating they do not act downstream of DSBs induced by TMPyP4. Together, these data show that dRTEL1 maintains male GSCs via different downstream events, including the expression of candidate genes and the DDR pathway.

## dRTEL1 regulates Stat92E level to prevent premature GSC loss

Thus far, our results demonstrate that dRTEL1 functions through several downstream targets to maintain male GSCs. To gain further understanding of the function of dRTEL1-mediated GSC maintenance, we examined the expression or activation of those known signaling pathways involving in GSC maintenance in *dRTEL1* mutant.

Dpp signaling activation promotes GSC maintenance and the signaling activation could be monitored by pMad expression. In controls, pMad was detected in GSCs but not in GBs (S6O Fig). Similarly, pMad was also detected in *dRTEL1* mutant GSCs (S6P Fig), suggesting that dRTEL1 likely does not function through the Dpp pathway for GSC maintenance.

The JAK/STAT signaling pathway is another signaling pathway to promote both GSC and CySC maintenance in testis [10,12]. *Stat92E*-deficient GSCs or CySCs were lost prematurely [10,12,66]. As reported previously, Stat92E was detected in GSCs, some GBs, and CySCs, but not in differentiated germ cells in control testes [13] (Fig 8A). We noted that the levels of

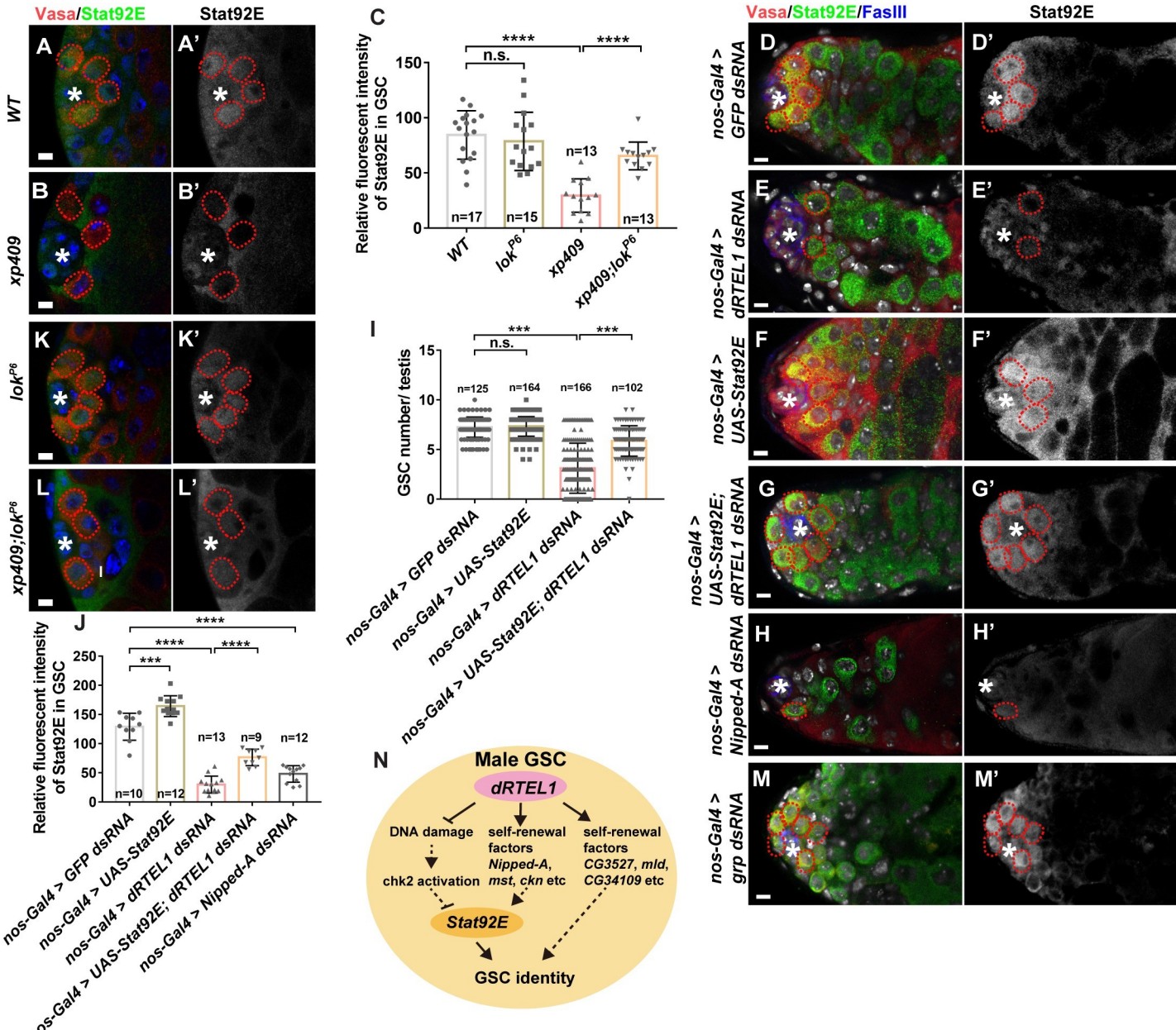

**Fig 8. dRTEL1 promotes Stat92E expression.** (A-B') Representative confocal images of larval testis at 72hr ALH showing Stat92E expression in WT (A,A') and *xp409* (B,B'). (C) Quantification of the relative fluorescent intensity of Stat92E in GSC in various backgrounds. Number in each bar represents the number of testes examined. (D-H') Representative confocal images of D14 adult testis showing Stat92E expression in *nos-Gal4 > GFP dsRNA* (D,D'), *nos-Gal4 > dRTEL1 dsRNA* (E,E'), *nos-Gal4 > UAS-Stat92E* (F,F'), *nos-Gal4 > UAS-Stat92E; dRTEL1 dsRNA* (G,G') or *nos-Gal4 > Nipped-A dsRNA* (H,H'). (I) Quantification of the GSC number per testis in various backgrounds. Number in each bar represents the number of testes examined. (J) Quantification of the relative fluorescent intensity of Stat92E per GSC in various backgrounds. Number in each bar represents the number of testes examined. (K-L') Representative confocal images of larval testis at 72hr ALH showing Stat92E expression in *lok^P6* (K,K') *and xp409;lok^P6* (L,L'). Data are mean ± s.e. n.s., not significant, *, P<0.05, **, P<0.01, ***, P<0.001. (M-M') Representative confocal images of D14 adult testis showing Stat92E expression in *nos-Gal4 > grp dsRNA*. (N) Schematic image showing the role of dRTEL1 in fly male GSCs. The hub is indicated by asterisks. GSCs are indicated by red dotted circles. DNA (TO-PRO-3) is in white in D-H and M and blue in A-B and K-L. Scale bars: 5 μm (A, B, and K-L) and 10 μm (D-H and M).

Stat92E protein decreased strongly in GSCs of *dRTEL1* larval testes (Fig 8B and 8C). Similarly, the Stat92E levels were significantly reduced in *nos-Gal4>UAS-dRTEL1 dsRNA* adult testes compared to *nos-Gal4>UAS-GFP dsRNA* controls (Fig 8D and 8E). It is worthy to note that

*Stat92E* transcription level was not altered in *dRTEL1* mutant testes (Fig 5C). We next wondered whether the reduced Stat92E expression in *dRTEL1* mutant contributes to the GSC loss and conducted genetic rescue experiment by ectopically expressing *Stat92E* using a *UAS--Stat92E* transgene driven by *nos-Gal4* in *dRTEL1* knockdown testes [67]. Similar to controls, adult testes with ectopic Stat92E expression sustained 6–9 GSCs (Fig 8F, 8I and 8J), which is consistent with previous report [13]. Notably, overexpression of Stat92E partially rescued GSC loss in *dRTEL1* knockdown testes (Fig 8D–8G, 8I and 8J). These testes harbored 5.9±0.2 GSCs (n = 102), compared with 3.1±0.2 GSCs (n = 166) in *dRTEL1* single knockdown testes. Together, these results suggest that dRTEL1 functions through Stat92E to regulate male GSC maintenance. Consistent with dRTEL1 acting through multiple downstream targets, testes with compromised activity of *Nipped-A*, *mst*, *ckn*, *tun*, or *CG7504* also exhibited reduced Stat92E expression (Figs 8H, 8J, and S6Q–S6T). In contrast, male GSCs of larvae or adult treated with TMPyP4 exhibited elevated DSBs but maintained Stat92E expression (S7 Fig), suggesting that induced DSBs do not affect Stat92E expression. Interestingly, Stat92E expression was partially restored in *dRTEL1* and *lok* double mutant GSCs (Fig 8C, 8K and 8L), suggesting that DNA damage checkpoint activation in *dRTEL1* mutant plays a role in Stat92E downregulation. Since compromising *CHK1* (*grp*) activity also partially suppressed GSC loss phenotype in *dRTEL1* mutant, it would be interesting to address whether it acts through Stat92E expression as well. In summary, these data indicate that dRTEL1 promotes maintenance of male GSCs via Stat92E (Fig 8N).

## dRTEL1 maintains female GSCs

We wondered whether *dRTEL1* also functions in female GSCs. The *Drosophila* ovarian GSCs are in direct contact with cap cells, contain an anteriorly-positioned spectrosome and express pMad (S8A Fig). Suggesting a role in ovarian GSCs, *dRTEL1* transcripts were detected in the germarium including GSCs (S8B Fig). Similarly, dRTEL1-GFP also expressed in ovarian GSCs (S8C–8E Fig). Next, we deployed the FLP/FRT system to generate marked female GSC clones and examined their maintenance during a course of three weeks after clone induction (ACI). While the percentage of control germaria carrying at least one marked GSC decreased slightly from 45% at day 3 ACI to 37% at day 21 ACI, the percentage of germaria containing *dRTEL1* mutant GSC clone dropped significantly from day 3 ACI to day 21 ACI (46% to 2% for *xp409* and 48% to 4% for *xp171*, S8F–8L Fig), indicative of defective GSC maintenance. Consistently, germaria with germline knockdown of *dRTEL1* contained few pMad positive GSCs, while germaria with somatic knockdown of *dRTEL1* harbored similar numbers of pMad-positive GSCs, similar to those of controls (S8M–S8Q Fig). Furthermore, *dRTEL1-GFP* transgene could fully rescue female lethality phenotype and these females exhibited normal germline development (S8S–S8W Fig). These data show that dRTEL1 activity is also required for the maintenance of ovarian GSCs.

As our early data suggest that dRTEL1 acts through multiple downstream targets to maintain male GSC, we investigated whether dRTEL1 also acts through these targets to regulate ovarian GSCs by examining the function of those candidate genes in the female germline. Similar to the controls, germaria with germline-specific knockdown of *mst*, *CG3527*, *mld*, *CG34109*, *tun*, *CG7504*, or *CG12926* contained 2–3 GSCs (S9A, S9C–S9E Fig and S5 Table). Surprisingly, germaria with germline knockdown of *Nipped-A* exhibited germline tumors filled with pMad-negative, spectrosome-containing undifferentiated germ cells (S9A and S9B Fig). Consistent with these results, germline-specific knockdown of *dRTEL1* in ovary did not result in transcriptional alteration of these genes (S9F Fig). These data suggest that dRTEL1 does not affect the expression of these candidate genes in female germline and likely acts through yet-to-be identified downstream target(s) for the maintenance of ovarian GSCs.

We next examined whether dRTEL1 also regulates female GSC maintenance via DNA damage response. In control germaria, γ-H2Av was detected in meiotic germline cysts but not in early germ cells including GSCs and CBs, while it accumulated in *dRTEL1*-deficient GSCs and CBs (S8X and S8Y Fig). Furthermore, p53 protein level was elevated and its activity reporter was activated in *dRTEL1* mutant GSCs (S9G–S9J Fig). These data suggest that dRTEL1 also functions to prevent activation of the DDR pathway in ovarian GSCs. Interestingly, further results show that ovary of females treated with TMPyP4 exhibited accumulation of DSBs but did not exhibit a GSC loss phenotype (S9K–S9L Fig), suggesting that dRTEL1 likely acts through additional targets to maintain ovarian GSCs. To test whether activation of the DDR pathway contributed to the observed GSC loss phenotype in *dRTEL1* mutant, we conducted genetic interaction for *dRTEL1* and downstream effector kinase *lok*. Surprisingly, *lok^{P6}* mutant did not prevent GSC loss in germaria with germline knockdown of *dRTEL1* (S9M Fig). This is different from an early report showing that GSC loss caused by induced DNA damage via I-CreI expression or X-ray irradiation was suppressed by *lok^{P6}* [50].

## Discussion

In this study, we identify the DNA helicase-like protein dRTEL1 as a novel regulator in the *Drosophila* male germline. Our data show that dRTEL1 acts cell-autonomously for GSC maintenance. Through transcriptome profiling and ChIP-Seq analyses, our results suggest that dRTEL1 might regulate the expression of multiple downstream targets required for GSC maintenance. In addition, the genetic interaction between *dRTEL1* and the DDR pathway components supports the notion that dRTEL1 involves in GSC maintenance partly via preventing DNA damage-induced checkpoint activation. Some of these downstream targets likely act through the expression of Stat92E, the key factor of male GSC maintenance, while others likely maintain GSC via yet-to-be-identified mechanism(s). Together, our findings have identified the important role of dRTEL1 in fly male germline stem cells (Fig 8N). Our results further show that dRTEL1 is also required for the maintenance of fly ovarian GSCs. As a conserved molecule ranging from insects (Drosophila and Mosquitoes) to vertebrate (human), RTEL1 might also play a role in stem cells in these diverse organisms.

### dRTEL1 affects the expression of intrinsic factors required for GSC maintenance

RTEL1 has two well-documented functions. First, RTEL1 maintains genome stability through promoting the disassembly of D-loop intermediates to prevent unwanted toxic DNA repair and stabilizes telomere integrity via dismantling T loops and removing telomeric DNA secondary structures [29,68]. Second, it suppresses formation of G4-DNA/R-Loops to prevent replication-transcription collisions and facilitate the fidelity of DNA replication [33–35]. *Drosophila* telomeres use a retrotransposon-dependent mechanism for their maintenance, different from most other eukaryotes whose telomeres are maintained by telomerase-based short repeats [69]. The underlying mechanism of fly telomere maintenance is different from that of other eukaryotes. Hence, it is likely that dRTEL1 does not have a function in fly telomeres. XPD, a paralog of RTEL1, is reported as a subunit of the transcription factor II H (TFIIH) and plays an essential role in transcription initiation [49]. Additionally, recent data show that RTEL1 plays a direct role in genome-wide replication via its interaction with proliferating cell nuclear antigen (PCNA) [40]. Unexpectedly, the PCNA interaction motif (PIP box) located at its C-terminus is not conserved in fly homolog, dRTEL1 (S1B Fig). Thus, the potential role of dRTEL1 in DNA replication awaits further investigation. Nevertheless, a functional dRTEL1-GFP reporter shows that dRTEL1 is a nuclear protein and our ChIP-Seq data show that dRTEL1 associates with chromatin and identify 654 dRTEL1 binding loci (Fig 5A).

Together with the publication showing that mammalian RTEL1 colocalizes with active RNAP II in transcriptional elongation, these data suggest that *Drosophila* RTEL1 might have a role in transcriptional regulation. Indeed, mouse MEFs exhibit extensive transcriptional changes upon loss of RTEL1 function. Our genome-wide transcriptome analyses also identify 1,817 genes which are affected by *dRTEL1* loss of function. However, it is worth noting that there is only minimal overlap between the transcriptome and ChIP-seq analyses, indicating that transcriptional alteration of most affected genes is likely an indirect consequence of DSBs induced by loss of dRTEL1 activity, as shown that mouse MEFs treated with G4/R-Loops inducer TMPyP4 exhibit a transcriptional change similar to that of RTEL1 loss of function [35].

Nevertheless, combining transcriptome profiling and ChIP-Seq analyses, we identify 71 candidate genes potentially regulated by dRTEL1. Among these candidates, 22 genes are positively regulated by dRTEL1 (S5 Table). Knocking down some of these genes including *Nipped-A*, *mst*, *CG3527*, *ckn*, *mld*, *CG34109*, *tun*, *CG7504* and *CG12926*, leads to a GSC loss phenotype in testis, reminiscent of *dRTEL1* mutant or germline knockdown phenotype, suggesting that they are novel players for male GSC maintenance. Additional data show that both *dRTEL1* and some of these genes act through Stat92E level to control GSC maintenance, supporting a role of dRTEL1 in regulating these candidate genes. Supporting this, the expression of these candidate genes is downregulated in the testes with germline-specific knockdown of *dRTEL1* and some of them are upregulated in the male germline upon dRTEL1 overexpression. Of note, the expression of these candidates was not altered in TMPyP4-treated S2 cells and male germline (in larval gonads or adult testes), showing that their expression is regulated by the DSB-independent activity of dRTEL1. Further research is needed to address how dRTEL1 affects the expression of these genes in male germline. Therefore, our findings show that dRTEL1 controls male GSC maintenance by promoting the expression of GSC intrinsic factors in addition to its role in preventing DNA damage checkpoint activation (see below).

The dRTEL1 activity is also required for the maintenance of ovarian GSCs. Female germline with defective *dRTEL1* activity exhibits a GSC loss phenotype. However, differing from their functions in testis, germaria with germline-specific knockdown of *mst*, *CG3527*, *mld*, *CG34109*, *tun*, *CG7504*, or *CG12926*, harbor 2–3 GSCs, similar to controls. Germaria with germline knockdown of *Nipped-A*, however, exhibit germ cell hyperplasia with ectopic pMad-negative, spectrosome-containing germ cells without GSC loss. In line with this, ovary with germline-specific knockdown of *dRTEL1* does not exhibit transcriptional alteration for these genes. These data suggest that dRTEL1 likely functions through some yet-to-be-identified downstream targets in female germline to regulate ovarian GSC maintenance. It is worthy to note that female GSCs defective in dRTEL1 function does not exhibit reduced pMad expression, suggesting that dRTEL1 might act downstream of or in parallel to the Dpp signaling pathway for the maintenance of ovarian GSCs.

## dRTEL1 maintains GSCs by preventing DNA damage-induced checkpoint activation

Previous studies have established a solid link between stem cell function and DNA damage response. It is reported that accumulated DNA damage is detected in aged stem cells and these DSBs are likely the cause of dysfunctional stem cells during ageing [70–72]. In this study, we show that, *dRTEL1* male GSCs exhibit elevated γ-H2Av signal and p53 activity, indicating the activation of the DDR pathway. Genetic data show that *ATR* is dispensable for the GSC loss in *dRTEL1* testis and germline knockdown of *ATM* is epistatic to germline-specific knockdown of *dRTEL1*. Furthermore, compromising either effector kinase (Chk1 or Chk2) partially prevents GSC loss in *dRTEL1* mutant. These data show that activation of the DDR pathway plays

a role in male GSC loss in *dRTEL1* mutant. Unexpectedly, TMPyP4 treatment induces elevated DSBs in both larval and adult GSCs, but does not lead to GSC loss. These data indicate that activation of the DDR pathway induced by dRTEL1 loss of function contributes to, but is not the main cause for the observed male GSC loss in *dRTEL1* mutant. Thus, dRTEL1 has a DSB-independent role in promoting maintenance of male GSCs.

In *Drosophila* female ovary, an early study shows that temporally-introduced DNA damage in GSCs leads to a rapid GSC loss by down-regulating BMP signaling and compromising GSC-niche adhesion via a CHK2-dependent pathway [50]. Another study also reports that Aubergine, a piRNA pathway component, functions in a similar CHK2-dependent pathway to control ovarian GSC self-renewal, in addition to its novel role in translational regulation [73]. Surprisingly, our results show that ovarian GSCs of TMPyP4-treated females exhibit ectopic DSBs but are maintained. Furthermore, ovarian GSCs with loss of *dRTEL1* function exhibit elevated activation of the DDR pathway and are lost, which is not prevented by *chk2* mutant that suppresses ovarian GSC loss induced by transient I-CreI expression or X-ray irradiation [50]. While the underlying mechanism remains unknown, one possible explanation is that the extent of DSBs induced by transient I-CreI expression or X-ray irradiation is different from (probably weaker than) those in *dRTEL1* GSCs. Furthermore, while induced DNA damage in female germline blocks GSC progeny differentiation, no blockage of progeny differentiation is observed in *dRTEL1* mutant or germline-specific knockdown ovary. This is also line with the notion that dRTEL1 acts through yet-to-be identified targets to promote ovarian GSC maintenance, in addition to suppress the activation of the DDR pathway.

### dRTEL1 functions through the JAK/STAT signaling pathway

The JAK/STAT pathway is involved in stem cell maintenance across a wide range of species. Dysfunction of JAK/STAT signaling is implicated in cancer and oncogenesis in mammals [74,75]. In this study, our results show that dRTEL1 regulates the JAK/STAT pathway in fly male germline. Loss of *dRTEL1* results in reduced Stat92E protein levels and precocious GSC loss, which could be partially prevented by forced Stat92E expression. Supporting this, GSCs with compromised downstream candidate targets of dRTEL1 exhibit a reduced Stat92E expression. However, dRTEL1 does not associate with the genomic region of *Stat92E* in our ChIP-Seq experiments. Furthermore, genome-wide profiling results also do not identify *Stat92E* as a candidate gene regulated by dRTEL1. So, how does dRTEL1 regulate Stat92E protein level? One possible explanation is that dRTEL1 functions through post-transcriptional machinery to control Stat92E protein level. Indeed, *CG7504*, one of the candidate targets, encodes a Forkhead-associated (FHA) domain-containing factor which has a putative function in termination of RNA polymerase II-mediated transcription [76]. Disrupting the function of *CG7504* may lead to transcription beyond termination site, thus producing abnormal and non-functional transcripts. However, this possibility awaits further investigation. Furthermore, we also cannot exclude other possibilities including translational regulation or post-translational degradation of Stat92E protein. Nevertheless, our data identify dRTEL1 as a novel regulator of the JAK/STAT signaling pathway. Interestingly, Stat92E protein levels are partially restored in *dRTEL1* mutant with compromised *Chk2* function, suggesting DNA damage response may play a role in the regulation of Stat92E protein level.

## Materials and methods

### *Drosophila* melanogaster

The fly stocks and crosses were raised on standard diet with 1% agar, 3.6% yeast, 2% yellow corn meal, 5.4% sugar and 3% molasses at 25˚C with a 12/12 light/dark cycle. Detailed

information of the fly stocks used in this study is provided here or FlyBase. The following Stocks were used: *w1118*, *y1w1118*, *FRT19A*, *nos-Gal4*, *tj-Gal4*, *hs-flp*, *UAS-p35(bl5073)*, *UAS-Stat92E* [67], *dRTEL1-GFP (v318751)*, *GFP RNAi (bl44415)*, *dRTEL1 RNAi (bl32975)*, *lok [p6]* (a gift from Toshie Kai), *Nipped-A (bl34849)*, *mst (bl29601)*, *CG3527 (v24762)*, *ckn (v24526)*, *mld (v101867)*, *CG34109 (v7657)*, *tun (v105713)*, *CG7504 (bl34683)*, *CG12926 (v32121)*, *lok* RNAi (*bl64482*), *xp157*, *xp171*, *xp409*, *UASp-dRTEL1-flag*.

*dRTEL1-flag* was subcloned into UASp.attB vector and transgenic lines were generated via target insertion utilizing attP landing site on III chromosome (Best Gene strain 9732) by Best-Gene, Inc. (ChinoHills, CA).

Knockdowns were performed using the *UAS/GAL4* system [77] by combining the *UAS-dsRNA* fly lines with cell-type specific GAL4 drivers and flies with desired genotype were maintained at 29˚C upon eclosure.

Female *dRTEL1* mutant clones were generated using (FLP)-mediated mitotic recombination technique [78]. Flies with genotype of *FRT19A. ubi-GFP; hs-flp* or *FRT19A.ubi-RFP; hs-flp* was crossed with flies with genotype *FRT19A.xp409*, *FRT19A.xp171*, or *FRT19A.xp157*. Females with the desired genotypes were selected to conduct time-course clonal analysis after clone induction (ACI). To generate marked GFP- or RFP-negative GSC clones, two-day-old females with desired genotypes were exposed to heat shock treatment for 1 hour at 37˚C, twice a day for 2 consecutive days. The heat shock-treated flies were maintained at 25˚C before dissection.

## Immunostaining and imaging

Testes or ovaries with desired genotype were dissected in PBS buffer or Shields and SangM3 insect medium (Sigma). The dissected samples were fixed in 4% PFA for 20 minutes at room temperature (RT). The samples were then incubated with primary antibodies diluted in 5% NGS for overnight at 4˚C. The rinsing and washing procedures were conducted and samples were then incubated with secondary antibodies diluted in PBT for 2 hrs at RT.

The following primary antibodies were used in this study: guinea pig anti-Vasa (1:10,000, a gift from Toshie Kai); rabbit anti-Zfh-1 (1:20,000, [79]); chicken anti-GFP (1:5,000, Abcam); rabbit anti-α-Spectrin (1:3,000, [80]); mouse anti-α-Spectrin (3A9, 1:200, monoclonal from Developmental Studies Hybridoma Bank (DSHB)); mouse anti-FasIII (7G10, 1:100, monoclonal from DSHB); rabbit anti-pMad (1:500, Cell Signaling); mouse anti-Flag (1:2,000, Sigma); rabbit-anti-Cleaved Caspase-3 (1:1000, ab208003, Abcam); mouse anti-γ-H2Av (UNC93-5.2.1, 1:2,000, monoclonal from DSHB); mouse anti-p53 (Dmp53 H3, 1:2,000, monoclonal from DSHB); guinea pig anti-Stat92E (1:1000, generated in our lab).

Secondary antibodies were obtained from Molecular Probes or Jackson ImmunoResearch Laboratories. Appropriate secondary antibodies conjugated with Alexa Fluor Cy3, Alexa Fluor 488, or Alexa Fluor Cy5 were used at a dilution of 1:500, 1:500, and 1:250, respectively. To-Pro-3 iodide or Hoechst (Invitrogen) was used to stain DNA at 1:10,000 and samples were subsequently mounted in Vectashield (Vector Laboratories). Images were obtained using Leica SP8 confocal microscope or Zeiss LSM 780 laser scanning confocal microscope and processed in Adobe Photoshop CC2018. The relative fluorescent Intensity of proteins in GSC were measured by FIJI (Image J).

## Fluorescent in situ hybridization (FISH)

Fluorescent in situ hybridization were performed as described previously [81]. Briefly, testes of ovaries were dissected in PBS and fixed in 4% PFA immediately at 4˚C overnight. Testis or ovary samples were treated with proteinase K (50 μg/ml in PBST; Sigma-Aldrich) for 5 min on

the second day. Samples were the refixed in 4% PFA, followed by prehybridization in hybridization solution (50% formamide, 5x SSC, 0.1% Tween-20, 50 μg/μl heparin, and 100 μg/ml salmon sperm DNA) for 1 h at 60˚C. Later, samples were hybridized with Dig-labeled probe overnight at 60˚C. After the samples were rinsed with PBST, they were incubated with anti-Dig-POD (1:200; Roche). The TSA Fluorescein system (Perkin Elmer) was used to develop in situ signals. The following primers were used to generate FISH probes: *dRTEL1*, 5'- CACCTA ATACGACTCACTATAGGGATGCCGGAGAGCCTGATCGCC -3' and 5'- CACCTATTTAGGTGA CACTATAGCTCGATGTTGTGGGCCTCATC -3'.

## TUNEL assay

TUNEL assay was conducted following the manufacturer's protocol provided by In Situ Cell Death Detection Kit (Roche). The ovary or testis samples were first fixed, permeabilized, and antibodies labeled. The samples were then washed thoroughly with PBS. The samples were then incubated in the TUNEL reaction mixture for 15–30 minutes at 37˚C with shaking a few times (avoiding light). The TUNEL labeled samples were washed for several times with PBT and proceeded to sample mounting and immunochemistry analysis.

## Propidium Iodide Nucleic Acid Stain

Propidium Iodide (PI, Thermo Fisher Scientific, 1:3,000) assay was conducted following the manufacturer's protocol provided by Propidium Iodide Nucleic Acid Stain (Invitrogen). For the positive control, the *hs-Gal4; UAS-hid* flies were heat shocked for 15 minutes and dissected 48 hours later. The fixed, permeabilized, and antibodies labeled testis samples were equilibrated in 2X SSC buffer (0.3M NaCl, 0.03M sodium citrate, pH 7.0). Testis samples were then incubated in 2X SSC with 100 μg/mL DNase-free RNase for around 20 minutes at 37˚C and then rinsed with 2X SSC for three times, 1 minute each. Dilute the 1 mg/mL Propidium Iodide stock solution at 1:3000 in 2X SSC to prepare a 500 nM solution of PI. Approximately 300 μL 500 nM solution of PI is capable of staining for one sample. The samples were further incubated with the diluted PI stain, for 1 to 5 minutes. The samples with PI incubation were then rinsed several times with 2X SSC. The PI labeled samples were washed for several times with PBT and proceeded to sample mounting and immunochemistry analysis.

## TMPyP4 treatment

To induce G4 stabilization, S2 cells were treated with various concentration (1μM, 10 μM and 50 μM) of TMPyP4 (Millipore, Cat#613560). For *Drosophila*, TMPyP4 was mixed with standard food to a final concentration of 1 μM, 10 μM and 50 μM to feed newly hatched L1 larvae till adult. The adult was continued fed with same concentration of TMPyP4 till dissection and examination.

## Quantitative real-time PCR

Extracted total RNA was used to synthesize cDNA through SuperScript III First-Strand Synthesis System (Invitrogen) according to the manufacturer's instructions. Oligo (dT)20 primer was used for cDNA library synthesis, which was then used for PCR or qPCR. KAPA SYBR FAST qPCR kit (KAPA Biosystems) was used to perform qPCR following the standard protocol provided by QuantStudio 5 Real-Time PCR System (Applied Biosystems). qPCR results were analyzed using QuantStudio 5 Real-Time PCR System. ΔCT from three independent biological replicates between different sample with specific time points or genotypes were shown

(mean ± s.e.). P-values and data significance was calculated according to two-tailed Student's t-test. Following primers used in the experiments were

*Adk1* forward (5'-CTGAGCAGCGGCGATTTGCTG-3'),

*Adk1* reverse (5'-GCCCGCGTGATGGCGTCGTTC-3'),

*Aub* forward (5'-CCGAGGGCGATCCGCGTGGC-3'),

*Aub* reverse (5'-CGACGCGGTACTGGTAGATG-3'),

*Cora* forward (5'-CCATTGATCGCAAAGCCATTG-3'),

*Cora* reverse (5'-GAGGCCAAGTGTCCGTGCGG-3'),

*CycE* forward (5'-CATTTAGTCGGGAGATGGCTT-3'),

*CycE* reverse (5'-GTTGCTGACTTGCTCATTCTG-3'),

*Egg* forward (5'-CTTGTATGAGGAGTATGCTGG-3'),

*Egg* reverse (5'-CTTCGATGAGCTGCAGCTTG-3'),

*Hid* forward (5'-CCACCCACTTCCCTCGAGCGC-3'),

*Hid* reverse (5'-GCGTGGCCGACGAGGTGGTGG-3'),

*HP1e* forward (5'-GGCGAGACCGTTTCGAATTTC-3'),

*HP1e* reverse (5'-CCCAGGTATTATCTTCATCGC-3'),

*Lis-1* forward (5'-GCGAGGAGCTTAACCAAGCG-3'),

*Lis-1* reverse (5'-CATCACTTTCTTCTGCAGCCG-3'),

*mei-41* forward (5'-CCAGATAGCAGCGAGTGCATC-3'),

*mei-41* reverse (5'-GAATATTCTTACAGTTATGGC-3'),

*nos* forward (5'-GGGCTGCACCTGCCACTGGG-3'),

*nos* reverse (5'-GTGGCCGCTGTTTGGGCCTGC-3'),

*Nup154* forward (5'-GTCTGCTGGAACTGACCGGCG-3'),

*Nup154* reverse (5'-CCAGAATCTCGTTGGGAATAG-3'),

*Rad1* forward (5'-GCTTTAATGATTATGGGATGG-3'),

*Rad1* reverse (5'-GACACTCGGACAGCACATTC-3'),

*Rga* forward (5'-ATGGCGAATTTAAATTTTCAAC-3'),

*Rga* reverse (5'-GTTGGCGAAGTCGGTTTGAAAC-3'),

*SkpE* forward (5'-CTTGAGTCCTCGGAAGGGGTG-3'),

*SkpE* reverse (5'-CTTGTGGTGATTGGCCCAGGC-3'),

*Su(var)205* forward (5'-CGACAGGCGGGTGCGCAAGG-3'),

*Su(var)205* reverse (5'-GGGTCGATCCTTCTTGGAGG-3'),

*U2A* forward (5'-GAGAACCTGGGCGCAACTCTG-3'),

*U2A* reverse (5'-GGAGCCCAGGTTGGGCACCG-3'),

*Zpg* forward (5'-GTGCCGACCAGATGCGGTTAG-3'),

*Zpg* reverse (5'-CATCGCACAAATGCTTCAGCC-3'),

*Stat92E* forward (5'- CGCCGAACGCAGAAGCTGCC-3'),

*Stat92E* reverse (5'- CTTGCGCTTGTTGGTGGCGC-3'),

*dRTEL1* forward (5'-GCTGTGCTCATCCTTGGCCTG-3'),

*dRTEL1* reverse (5'-CGCACGACCCATGGTCTTGGC-3'),

*mst* forward (5'-CCCCAAGCGGATACCCGCCTG-3'),

*mst* reverse (5'-ATACCGTCTCCCACTGGAGTC-3'),

*Nipped-a* forward (5'-CAATGCAACACATACGTAAACTG-3'),

*Nipped-a* reverse (5'-GATGGACGAAAATGCTTATGCAG-3'),

*CG3527* forward (5'-GGAGACGGTGAAGGTGCACAAC-3'),

*CG3527* reverse (5'-CTCGGTGCGCACAAAGACCTG-3'),

*ckn* forward (5'-GATCCATAGAGGAATGGCTGC-3'),

*ckn* reverse (5'-GTCCTTCACACGCTTGATGGC-3'),

*mld* forward (5'-GCGACGATGGACTGCCGCAAG-3'),

*mld* reverse (5'-GACGCTTCGCTTGTCTGGCCG-3'),

*CG34109* forward (5'-CACCTCAAATAATGCCCTGCTG-3'),

*CG34109* reverse (5'-GCGTTGTTATTCTCCGGAAG-3'),

*tun* forward (5'-CGAGGAGAACGTGTGGAAGC-3'),

*tun* reverse (5'-CCCATATCACAACTTGATCGTC-3'),

*CG12926* forward (5'-GCCTCCTCCGATTGCCGCAGC-3'),

*CG12926* reverse (5'-GCATTGTCGTCTTCCCTTATC-3'),

*CG7504* forward (5'-CTGAATGGGCTCTTCATCAAC-3'),

*CG7504* reverse (5'-GATTTCCTCGACTACCGGCTC-3'),

*Actin5C* forward (5'-GGATCTCCAAGCAGGAGTACG-3'),

*Actin5C* reverse (5'-TCCTCCAGCAGAATCAAGACC-3'),

*Rp49* forward (5'-CAATCCTCGTTGGCACTCACC-3'),

*Rp49* reverse (5'-TCCGCCCAGCATACAGGC-3'),

*RNA polymerase II* forward (5'-TGAGAGATCTCCTCGGCATTCT-3'),

*RNA polymerase II* reverse (5'-ACTGAAATCATGATGTACGACAACGA-3').

## Quantification and statistical analysis

Number of GSCs or spectrosome-containing cells were counted from randomly selected testes or ovaries under a fluorescence microscope and every GSC was carefully assessed by adjusting

the focus to avoid those hidden GSCs. The number of GSCs in testis or ovary was counted according to Vasa staining, α-Spectrin staining and the position (GSCs are attached to hub cells in testis or cap cells in ovary). For ovary, the percentage of germaria carrying a marked GSC clone was calculated by dividing the number of germaria carrying a GFP-negative marked GSC clone with the total number of germaria checked. The relative fluorescent intensity of Stat92E, γ-H2Av and p53 in GSCs was measured by Image J. Data processing was analyzed and performed using GraphPad Prism 7.0 (GraphPad Software Inc.). P values were determined by two-tailed Student's t tests. $P < 0.05$ represents significant statistical difference, error bars indicated standard deviation.

### RNA-seq and data analysis

Total RNA was extracted using TRIzol reagent following the manufacturer's instructions. 40 pairs of $w^{1118}$ or *dRTEL1* or *dRTEL1; dRTEL1-GFP* early L3 (60-72hr) stage testes were dissected in Schneider's medium (Invitrogen) and used as one set of data respectively. For RNA-Seq, each genotype was sequenced with two replicates. The integrity of RNA was confirmed by gel electrophoresis. Subsequent mRNA purification, library construction, sequencing and data analysis were performed by BGI (Beijing Genomics Institute, Shenzhen, Guangdong, China). In brief, TruSeq RNA V2/Illumina kit was used to generate the Illumina cDNA libraries. Libraries were sequenced with Illumina HiSeq 2500. Raw sequencing reads were cleaned by removing adaptor sequences, reads containing polyN sequences, and low-quality reads. The *Drosophila* genome (dm6, FlyBase 6.05) was used to align and filter reads. After the sequencing reads were aligned, normalization was performed and then FPKM (fragments per kilobase per million mapped reads) was calculated using a software package called RSEM. The FDR (false discovery rate) $< 0.01$ and the absolute value of log2 Ratio $\geq 2$ were used to identify differentially expressed genes (DEGs) in $w^{1118}$ versus *dRTEL1* and $w^{1118}$ versus *dRTEL1; dRTEL1-GFP* samples. Annotation analysis of Gene Ontology (GO) was performed for identified DEGs. GO enrichment analysis was performed on www.flymine.org/ with HolmBonferroni correction with a maximum P-value of 0.05.

The RNA-seq generated during this study are available at Mendeley Data (https://data.mendeley.com/datasets/9x27fk49nm/1, https://data.mendeley.com/datasets/8zgxnk8fg8/1, https://data.mendeley.com/datasets/bs72w8ynz5/1).

### Chromatin immunoprecipitation sequencing (ChIP-Seq) and data analysis

Newly enclosed *Drosophila* males with dRTEL1-GFP expression were selected and aged for 3 days at 25˚C. 800 pairs of adult testes were collected for each ChIP-seq reaction as one replicate. Two replicates were generated for control or experiment group. The whole ChIP process was conducted at 4˚C or on ice. ChIP-IT high sensitivity kit (Active motif) was used to perform the ChIP experiments according to the manufacturer's instruction. Sonication was conducted on Bioruptor sonicator (diagenode) for 30 second on-off pulse, 30 cycles at 4˚C. The fragment size of sonicated chromatin was confirmed between 200bp to 500bp. The chromatin was immune-precipitated by a ChIP-grade GFP antibody (ab290, Abcam) and a ChIP-grade IgG antibody (ab171870, Abcam).

Illumina TruSeq ChIP Sample Preparation Kit (IP-202-1012) was used to generate the ChIP-seq libraries following the manufacturer's instruction. The Illumina compatible libraries were sequenced with Mi-seq desktop sequencer (Mi-Seq, Illumina) by BGI, China. 75 bp single-end read sequencing was accomplished. After sequencing data was delivered, data filtering including removing adaptor sequences, contamination and low-quality reads from raw reads was conducted. After filtering, the remaining reads were called "clean reads", which were

mapped to Drosophila genome (dm6, FlyBase 6.05). Then the alignment results were used to calling peak. The model-based analysis of ChIP-Seq (MACS) peak-finding algorithm version 1.4.1 was used for peak calling. The candidate Peak region was extended to be long enough for modeling. Dynamic Possion Distribution was used to calculated p-value of the specific region based on the unique mapped reads. The region would be defined as a Peak when p-value < le-05. Later, the peak information was used for standard bioinformatics analysis. MAnorm was used to identify differential peaks between samples. The log2 ratio of read density between samples M was plotted against the average log2 read density A for all peaks, and robust linear regression was applied to fit the global dependence between the M-A values of common peaks. Then the derived linear model was used as a reference for normalization and extra polated toall peaks. Finally, the P-value for each Peak was calculated based on Bayesian model, the significant regions were picked up if $|M| >= 1$ and p-value$<= 10-5$. Peaks were classified based on the location (UCSC annotation data) and showed in the following genome regions: intergenic, introns, downstream, upstream and exons.

The CHIP-seq Data generated during this study are available at Mendeley Data (https://data.mendeley.com/datasets/vyzccs8tsh/1).

## Supporting information

**S1 Fig. Identification of GSC loss phenotype in *dRTEL1* flies.** (A) scheme showing EMS-based mutagenesis screening on X-chromosome. FRT19A male flies treated with 25mM EMS and crossed with an X chromosome balancer lethal(1)/Fm7.Kr.GFP to generate about 2000 stocks. After screening these lines, 4 mutant lines identified with defects in both female and male germlines. (B) The protein structure of RTEL1 and the molecular lesion present in the *dRTEL1* mutant alleles. *Drosophila*, human, and *M. musculus* RTEL1 proteins contain a DEAD_2 (yellow), a Helicase_C_2 (blue) domain and HN_RTEL1 domain (orange), while human, and *M. musculus* RTEL1 proteins contain an extra PIP box (green). d, *Drosophila melanogaster*; Hs, *Homo sapiens*; Mm, *Mus musculus*. (C) Verification of corresponding mutation in *xp409*, *xp171* and *xp157* by RT-PCR followed by sequencing. (D-G") *dRTEL1–GFP* (D-D") transgene fully rescues germline defect observed in *xp409* (E-E"), *xp171* (F-F") and *xp157* (G-G") larval gonad at 96 hr ALH. (H) Quantification of the GSC number per testis in various backgrounds. Number in each bar represents the number of testes examined. Data are mean ± s.e. n.s., not significant, *, P<0.05, **, P<0.01, ***, P<0.001. The hub is indicated by asterisks. GSCs are indicated by red dotted circles. DNA (TO-PRO-3) is in white in D-G. Scale bar 5 um.
(TIF)

**S2 Fig. dRTEL1 is required cell-autonomously for maintenance of male GSCs.** (A) *dRTEL1* sense probe detecting no specific signal in WT testis at 96 hr ALH. (B) A WT testis at 96 hr ALH showing *dRTEL1* transcripts detected by anti-sense probe (arrows). (C-D') *xp409* testis at 48 hr ALH (D,D') exhibiting a decrease of somatic cell number compared with WT counterparts (C,C'). (E) Quantification of the Zfh-1 positive cell number per testis in various backgrounds. Number in each bar represents the number of testes examined. (F-I) Representative images of 96 hr ALH testis of WT (F), *tj-Gal4 > UAS-dRTEL1-flag* (G), *xp409* (H) and *xp409*; *tj-Gal4 > UAS-dRTEL1-flag* (I). (J) Quantification of GSC number per testis in various backgrounds. Number in each bar represents the number of testes examined. Data are mean ± s.e. n.s., not significant, *, P<0.05, **, P<0.01, ***, P<0.001. The hub is indicated by asterisks. GSCs are indicated by red dotted circles. DNA(TO-PRO-3) is in white. Scale bar: 5 μm.
(TIF)

**S3 Fig. Summary of the RNA-seq data.** (A-B') Representative confocal images of GSCs in control (A,A') and *xp409* (B.B') larval testis at 60–72 hr ALH. ToPro-3 in blue. Scale bar: 10 μm. (C) The differentially expressed genes categorized according to gene molecular function ontology term analysis. (D) The differentially expressed genes categorized according to gene biological process ontology term analysis. (E) The randomly selected genes exhibiting same trend of expression change (by log2 fold) in $w^{1118}$ testis and *dRTEL1* mutant testis in the RNA-Seq (N = 2) and qRT-PCR (N = 3), although the exact fold changes in transcription levels showing some variations. (F) The comparisons of RNA-seq data with 2 published RNAi screens.
(TIF)

**S4 Fig. Summary of the ChIP-Seq data.** (A) The protein domain structure of dRTEL1 and XPD. (B) GO term (biological process) analysis for genes enriched in *dRTEL1* mutant. (C) GO term (molecular function) analysis for genes enriched in *dRTEL1* mutant. (D) Summary of a small-scale RNAi screen on the 22 down-regulated overlapping genes in the fly testis. (E) Relative mRNA levels of *Nipped-A*, *mst*, *CG3527*, *ckn*, *mld*, *CG34109*, *tun*, *CG7504*, and *CG12926* in *dRTEL1* germline knockdown testis. (F) Relative mRNA levels of *Nipped-A*, *mst*, *CG3527*, *ckn*, *mld*, *CG34109*, *tun*, *CG7504*, and *CG12926* in testis with germline overexpression of *UASp-dRTEL1-flag* in combination with *nos-Gal4*. (G) Relative mRNA levels of *Nipped-A*, *mst*, *CG3527*, *ckn*, *mld*, *CG34109*, *tun*, *CG7504*, and *CG12926* in S2 cells with dsRNA-mediated *dRTEL1* knockdown. Data are mean±s.e. n.s., not significant, *, P<0.05, **, P<0.01, ***, P<0.001.
(TIF)

**S5 Fig. The effect of TMPyP4 on expression of candidate target genes.** (A) Relative mRNA levels of *Nipped-A*, *mst*, *CG3527*, *ckn*, *mld*, *CG34109*, *tun*, *CG7504*, and *CG12926* in S2 cells treated with various concentration of TMPyP4 (1 μM,10 μM or 50 μM). (B) Relative mRNA levels of *Nipped-A*, *mst*, *CG3527*, *ckn*, *mld*, *CG34109*, *tun*, *CG7504*, and *CG12926* in L3 larva testis treated with various concentration of TMPyP4 (1 μM,10 μM or 50 μM) from hatching. (C) Relative mRNA levels of *Nipped-A*, *mst*, *CG3527*, *ckn*, *mld*, *CG34109*, *tun*, *CG7504*, and *CG12926* in D14 adult testis treated with various concentration of TMPyP4 (1 μM,10 μM or 50 μM) from larval hatching. Data are mean±s.e. n.s., not significant, *, P<0.05, **, P<0.01, ***, P<0.001.
(TIF)

**S6 Fig. Identification of DSB accumulation in *dRTEL1* testis.** (A,A') Representative image of *p53R-GFPcyt* larval testis at 72hr ALH showing no GFP expression in GSCs. (B,B') A *xp409; p53R-GFPcyt* larval testis at 72hr ALH showing GFP expression in GSCs. (C,C') Representative image of D14 WT testis. (D,D') Representative image of D14 *nos-Gal4 > GFP dsRNA* testis. (E,E') Representative image of D14 *nos-Gal4 > dRTEL1 dsRNA* testis exhibiting GSC loss. (F-G') Representative image of D1 *nos-Gal4 > tefu dsRNA* testis (F,F') or D1 *nos-Gal4 > tefu dsRNA*, *dRTEL1 dsRNA* testis (G,G') showing complete GSC loss. (H) Quantification of GSC number per testis in WT and *lok^{P6}*. Number in each bar represents the sample number. (I) Quantification of GSC number per testis in various background. Number in each bar represents the number of testes examined. Data are mean±s.e. n.s., not significant, *, P<0.05, **, P<0.01, ***, P<0.001. (J-N') Representative confocal images of D14 testis showing γ-H2Av expression in *nos-Gal4 > GFP dsRNA* (J,J'), *nos-Gal4 > Nipped-A dsRNA* (K,K'), *nos-Gal4 > CG3527 dsRNA* (L,L'), *nos-Gal4 > mst dsRNA* (M,M'), and *nos-Gal4 > CG7504 dsRNA* (N,N'). (O,O') A WT larval testis at 72 hr ALH showing pMad expression in GSCs. (P, P') A *xp409* larval testis at 72 hr ALH showing pMad expression in GSCs. (Q-T')

Representative confocal images of D14 testis showing Stat92E expression in *nos-Gal4 > mst dsRNA* (Q,Q'), *nos-Gal4 > ckn dsRNA* (R,R'), *nos-Gal4 > tun dsRNA* (S,S'), or *nos-Gal4 > CG7504 dsRNA* (T,T'). The hub is indicated by asterisks. GSCs are indicated by red dotted circles. DNA(TO-PRO-3) is in white in A-G and O-T and blue in J-N. Scale bars: 5 μm. (TIF)

**S7 Fig. Male GSCs of larvae or adult treated with TMPyP4 maintain Stat92E level.** (A) Stat92E expression in WT larval testis (96 hr ALH) treated with 0, 1μM, 10μM and 50 μM TMPyP4. (B) Stat92E expression in WT testis (D14) treated with 0, 1μM, 10μM and 50 μM TMPyP4. (C) Quantification of the relative fluorescent intensity of Stat92E per GSC in larval testis of various backgrounds at 96 hr ALH. (D) Quantification of the relative fluorescent intensity of Stat92E per GSC in various backgrounds at D14. Data are mean±s.e. n.s., not significant, *, $P<0.05$, **, $P<0.01$, ***, $P<0.001$. The hub is indicated by asterisks. GSCs are indicated by red dotted circles. DNA(TO-PRO-3) is in blue in A and white in B. Scale bars: 5 μm. (TIF)

**S8 Fig. dRTEL1 is required for the maintenance of ovarian GSCs.** (A) A schematic diagram showing the anterior half of the *Drosophila* germarium. (B,B') Representative image of D14 WT germarium showing *dRTEL1* transcripts in GSCs detected by anti-sense probe (arrows). (C,C') Representative image of D14 *dRTEL1-GFP* germarium showing that GFP expression in germ cells. (D,D') Representative image of D14 *dRTEL1-GFP;nos-Gal4 > dRTEL1 dsRNA* germarium showing reduced GFP expression in germ cells. (E,E') Representative image of D14 *dRTEL1-GFP;tj-Gal4 > dRTEL1 dsRNA* germarium showing reduced GFP expression in Tj-positive somatic cells. (F-K) Representative images showing a marked GFP-negative GSC clone in control (F,G), *xp409* (H,I) or *xp171* (J,K) D7 or D21 ACI. Note that D21 *xp409* or *xp171* germarium do not contain marked GFP-negative GSC clone. (L) Quantification of the percentage of germaria with marked GFP-negative GSC clones in various backgrounds. Number in each bar represents the sample number. (M-P) Representative images of D14 germarium of *nos-Gal4 > UAS-GFP dsRNA* (M-M'), *nos-Gal4 > UAS-dRTEL1 dsRNA* (N-N"), *tj-Gal4 > UAS-GFP dsRNA* (O,O') or *tj-Gal4 > UAS-dRTEL1 dsRNA* (P, P') showing pMad-positive GSCs. (Q) Quantification of pMad positive cells per germarium in H-I. Number in each bar represents the number of testes examined. (R-T) Representative images of ALH 96hr germarium of *WT* (R), *dRTEL1-GFP* (S), or *dRTEL1; dRTEL1-GFP* (T). (U-W) Representative images of D14 germarium of *WT* (U-U'), *dRTEL1-GFP* (V-V'), or *dRTEL1; dRTEL1-GFP* (W-W'). (X-X') A D14 germarium showing no γ-H2Av detected in marked GFP-negative WT GSC. (Y-Y') A D14 germarium showing γ-H2Av accumulation in marked GFP-negative *dRTEL1* GSC. Data are mean±s.e. n.s., not significant, *, $P<0.05$, **, $P<0.01$, ***, $P<0.001$. GSCs are indicated by red dotted circles. DNA (TO-PRO-3) is in white in B and M-O and blue in C-K and R-Y. Scale bar: 10 μm. (TIF)

**S9 Fig. dRTEL1 maintains ovarian GSCs via yet-to-be identified downstream targets.** (A, A') A representative image of D14 *nos-Gal4 > UAS-GFP dsRNA* germarium. (B,B') A representative image of D14 *nos-Gal4 > Nipped-A dsRNA* germarium exhibiting germline tumors filled with pMad-negative, spectrosome-containing undifferentiated germ cells. (C,C') Representative image of D14 *nos-Gal4 > CG3527 dsRNA* germarium containing 2 GSCs. (D,D') Representative image of D14 *nos-Gal4 > CG7504 dsRNA* germarium containing 2 GSCs. (E) Relative mRNA levels of *Nipped-A*, *mst*, *CG3527*, *ckn*, *mld*, *CG34109*, *tun*, *CG7504*, and *CG12926* in *dRTEL1* germline knockdown ovary. (F) Quantification of the knockdown efficiency in ovary by qPCR. (G-G") A D14 marked RFP-negative WT GSC does not express

p53R-GFPnls. (H-H") A D14 marked RFP-negative *xp409* GSC expressing p53R-GFPnls. (I-I')
A D14 marked GFP-negative WT GSC showing no p53 expression. (J-J') A D14 marked GFP-
negative *xp409* GSC showing elevated p53 expression. (K) γ-H2Av and p53 expression in D14
WT ovary treated with 0, 1μM, 10μM and 50 μM TMPyP4. (L) Quantification of the relative
fluorescent intensity of γ-H2Av and p53 in GSCs and number of GSCs per ovary in various
backgrounds at D14. (M) Quantification of pMad positive cells per germarium in various
backgrounds. Number in each bar represents the number of testes examined. Data are
mean ± s.e. n.s., not significant, *, P<0.05, **, P<0.01, ***, P<0.001. (L-L') DNA (TO-PRO-3)
is in white in A-D and G-H and blue in I-K. GSCs are indicated by red dotted circles. Scale
bars: 10 μm.
(TIF)

**S1 Table. Genes upregulated in *dRTEL1* testis.**
(XLSX)

**S2 Table. Genes downregulated in *dRTEL1* testis.**
(XLSX)

**S3 Table. Genes identified to be required for germline development in other studies.**
(XLSX)

**S4 Table. The list of genes differentially expressed between *dRTEL1* testes and wild-type
control testes and enriched by *dRTEL1* at genomic regions.**
(XLSX)

**S5 Table. Small scale RNAi screening results of candidate genes.**
(XLSX)

## Acknowledgments

The authors thank DSHB, BDSC (NIH P40OD018537), VDRC and TRiP at Harvard Medical
School (NIH/NIG R01-GN08494) for reagents; the Temasek Life Sciences Laboratory confocal
facility and sequencing facility for supports; and Dr Tong-Wey Koh, Dr Gyeong Hun Baeg
and the members of the Y.C. laboratory for discussion.

## Author Contributions

**Conceptualization:** Ying Yang, Yu Cai.

**Data curation:** Ying Yang, Yu Cai.

**Formal analysis:** Ying Yang.

**Funding acquisition:** Ying Yang, Zhouhua Li, Yu Cai.

**Investigation:** Ying Yang, Ruiyan Kong, Feng Guang Goh, W. Gregory Somers, Gary R.
Hime.

**Methodology:** Ying Yang.

**Project administration:** Yu Cai.

**Resources:** Ying Yang, Gary R. Hime, Zhouhua Li, Yu Cai.

**Software:** Ying Yang, Yu Cai.

**Supervision:** Yu Cai.

**Validation:** Ying Yang, Yu Cai.

**Visualization:** Ying Yang, Yu Cai.

**Writing – original draft:** Ying Yang.

**Writing – review & editing:** Ying Yang, Gary R. Hime, Zhouhua Li, Yu Cai.

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
