## [Decision Letter · Decision Letter 0]

13 May 2021

Dear Dr Cai,

Thank you very much for submitting your Research Article entitled 'dRTEL1 Is Essential for the Maintenance of Drosophila male Germline Stem Cells' to PLOS Genetics. I apologize for the length of time it has taken to obtain reviews.

The manuscript was fully evaluated at the editorial level and by three independent peer reviewers. The reviewers appreciated the attention to an important problem, but raised some substantial concerns about the current manuscript. Based on the reviews, we will not be able to accept this version of the manuscript, but we would be willing to review a much-revised version. We cannot, of course, promise publication at that time.

Should you decide to revise the manuscript for further consideration here, your revisions should address the specific points made by each reviewer. We will also require a detailed list of your responses to the review comments and a description of the changes you have made in the manuscript. Many of the comments of Reviewers 1 and 2 can be addressed through additional writing, but please pay particular attention to point #6 of Reviewer 1 and points #3, #8, and #11 of Reviewer 2. Reviewer 3 had more fundamental concerns, particularly whether your data exclude the possibility that the defects you see are all due to DNA repair defects. This is an important point, since this is a major function identified through research in other organisms. Going from the fact that RTEL1 has a Rad3/XPD helicase domain to suggest it may regulate gene expression is a bit of a jump, given that XPD participated in transcription, not gene regulation per se. I find it troubling that most of the ChIP-seq signal is along gene bodies and the overlap between genes identified in this experiment versus in RNAseq is relatively low. I believe this argument needs to be strengthened to be convincing evidence for a new function. At a minimum, can you cite other examples of similar ChIP-seq profiles (*e.g.*, mostly gene body) for proteins understood to be transcriptional regulators?

If you decide to revise the manuscript for further consideration at PLOS Genetics, please aim to resubmit within the next 60 days, unless it will take extra time to address the concerns of the reviewers, in which case we would appreciate an expected resubmission date by email to plosgenetics@plos.org.

[LINK]

We are sorry that we cannot be more positive about your manuscript at this stage. Please do not hesitate to contact us if you have any concerns or questions.

Yours sincerely,

Jeff Sekelsky, PhD

Guest Editor

PLOS Genetics

Gregory P. Copenhaver

Editor-in-Chief

PLOS Genetics

Reviewer's Responses to Questions

**Comments to the Authors:**

**Reviewer #1**: Dear Authors,

Yang et al manuscript titled ‘dRTEL1 Is Essential for the Maintenance of Drosophila male Germline Stem Cells’ is a very compelling and comprehensive story indicating a novel function of Drosophila RTEL1 as a regulator of GSCs. This article, once published, will be of great interest to the PLOS Genetics audience.

The authors utilized a number of different techniques to address their research questions. In the paper we can see the mastery of confocal imaging and traditional genetic approaches like gene knockdown and rescue, complemented by big data analyses (ChIP-seq and RNA-seq). RTEL1 helicase, traditionally implicated in telomere maintenance and DNA repair, as demonstrated here, does seem to play a critical role in germline stem cell maintenance. The authors thoroughly investigate RTEL1’s role in Drosophila testis GSCs, however a brief look at the female germline indicates some elements of this regulation are conserved which present an opportunity for a follow-up publication.

I did not find any major issues preventing the publication of this manuscript (perhaps with the exception of a few missing controls). However, there is a number of minor revisions the authors should address to help the Plos Genetics audience fully appreciate the scope, quality, and significance of this paper:

Data presentation:

1) To facilitate the identification of GSCs, the authors had an excellent idea to provide a schematic of testis in Figure 1A and mark the hub cells with an asterisk throughout the paper. For the benefit of the audience that is not versed in the testis topology, I would strongly suggest outlining GSCs (and other cells which the phenotype the authors discuss or compare) in all images. The authors already demonstrated the importance of this approach in Fig 8M-N where the cells of interest are outlined. Figure 8 could also benefit from a germarium schematic alike the one seen in Figure 1A. In figures discussing subcellular structures e.g., Figure 3K, arrows could be used in a similar fashion to how the data is presented in Figure 3L, which would improve the reading experience.

2) In line 134 the authors explain the significance of α-Spectrin and Vasa stains. With the exception of Figure 4, all figures contain images of cells stained with either Zfh-1, pMad, or FasIII. Zfh-1 is mentioned once in line 166, but the significance and the role of this stain needs to be elucidated. Similarly, pMad is mentioned for the first time in line 355 in the context of Figure 7. However, the readers could benefit from knowing the significance of this stain much earlier in the story since pMad is labeled in Figures 2+. I did not find any mention of FasIII in the manuscript. The information regarding FasIII should be added to the Materials and methods section (line 528) as well. The schematic in Figure 1A presents an excellent opportunity to introduce all these stains to the audience as well.

3) The authors in line 131 mention ‘a typical Drosophila larval testis contains 6 to 9 GSCs adjacent to the hub located at the apical tip of the testis’. The schematics in Figure 1A contains 3 GSC cells while the average GSC count per testis in Figure 1H is almost 8. Does the manuscript contain any images where more than 5-6 GSCs are present? Perhaps a clarification is needed in the ‘Quantification and Statistical analysis’ (line 681), whether the missing GSCs are on the distal side of the testis, therefore hidden from sight? Should it be the case, how are these quantified? Do the images used for quantification look like Figure 3A or 3K, and the lateral view images like Figure 3B, 3L, 3M are excluded?

4) In the section concerning female GSCs (line 385), the authors indicated they are counting the GFP-marked GSCs (line392-394). The audience could benefit from a note on how these cells were marked. Additionally, In Figure 8G the authors perform a quantification of the % of germaria carrying a marked GSC clone. Figure 8A needs a series of images that depict both GFP-marked and GFP-lost cells to convey how do the quantified phenotypes look like with the cells of interest outlined. Additionally, the Figure 8 legend (lines 1057-1058) contains a reference to the RFP signal. The audience could benefit from a thorough description of how the RFP signal came into play and its significance.

Experimentation and controls:

5) The authors developed and described the nature of 3 dRTEL1 mutants xp157, xp171 and xp409 (lines 113-130). Traditionally, a Western Blot approach would be utilized to inform whether the protein of interest is being produced (either full-length, a truncated form, or no protein at all). It would be too much to ask to conduct such analysis given the lack of commercially available dRTEL1 antibodies, however, these mutants could be characterized further at the mRNA level to confirm the sequence-based predictions.

6) In Figure 3I-J, the authors performed a propidium iodide staining to infer regarding cell-death in dRTEL1 deficient cells. Negative results without a positive control are impossible to interpret. Are the cells propidium iodide negative because they have an intact membrane, the imaging conditions were suboptimal, or a different reagent was accidentally used? The authors should include a condition in which the audience of the paper can see how propidium iodide-positive testis cells look like. Similarly Figure 3A-B requires a positive control for Caspase-3 staining and Figure 3C-D requires a positive control for TUNEL (these images could become a part of the supplementary material but are necessary for proper data interpretation). It is worth noting that all xp409 images, unlike WT, present a weak signal in these 3 assays. Is it a background or is it real? The interpretation of this section could change depending on what the positive controls look like.

7) Line 364 reads ‘We noted that the levels of Stat92E protein decreased strongly in dRTEL1 larval testes (Fig.7B).’ Quantification and clarification might be needed regarding what constitutes a ‘strong decrease’. Given the signal spans the entire testis, what is the area of interest? Since dRTEL1 mutants exhibit diminished number of GSCs could the loss of Stat92E signal be attributed to the loss of GSCs instead?

**Reviewer #2**: The manuscript entitled as dRTEL1 Is Essential for the Maintenance of Drosophila male Germline Stem Cells by Yang et al. describes function of CG4078. CG4078 encodes a fly homolog of Regulator of Telomere Elongation Helicase and paralogue of XPD. Though dRTEL is expressed in germ and somatic cells, the only germline expression contributes GSCs maintenance cell-autonomously in males. GSCs loss is not due to apoptosis but precocious GSC differentiation. ChIP and genetic analyses identified 9 functional regulated genes including Nipped-A those cause GSC loss. Curiously, GSCs maintenance by dRTEL was independent of Dpp pathway but dependent on the JAK-STAT pathway. dRTEL1 was also required in the female germline to maintain ovarian GSCs to prevent the activation of the DDR. However, the downstream genes and molecular mechanisms are yet to be identified. Though dRTEL’s primary and/or direct target(s) triggering precocious differentiation resulting in GSC loss remains elusive, comprehensive analyses and convincingly demonstrated the important role of dRTEL. The contents of this study are suitable to be published in PLoS genetics. The authors should address concerns and points described below, making the results more convincing and figures appropriately organized. Especially, the analyses of mRNA-seq and ChIP-seq should be described carefully in the main text and method section. Second rounds of revision would be necessary.

Major points:

1. As to manuscript organization: mutant alleles and knockdown animals at different stages, such as larva and adult, were examined in each experiment. To avoid readers’ confusion, the stage of the examined samples should be clearly described in both main figs and supplemental figs.

2. L208-218: dRTEL1 mutant testes exhibited precocious differentiation of germline cells. It would be more convincing if the authors identify spermatogonia by any specific markers, such as Stat92E, in addition to spectrin staining.

3. Line 219: the mRNA-seq data may reflect the loss of germline cells in dRTEL1, not the change of mRNA levels, because the GSCs were lost in mutant ALH 96 hr testes. ALH 60-72 hr testes were used as the RNA-seq samples without any explanation. The GSC numbers and the staining images should be shown at ALH 60-72 hr, or this concern should be described in the main text.

4. Line 237: ‘Among those 986 down-regulated genes in dRTEL1 testes, 136 genes were previously identified…’ Is there any reason to focus on the down-regulated genes? Among 831 up-regulated genes in dRTEL1 testes, how many genes are included in the previous studies cited here?

5. Line 242: ‘These data show that dRTEL1 regulates genes required for GSC maintenance, directly or indirectly.’ This sentence may be overemphasized. There are only 20% overlap between the listed genes in this study and those in the previous studies. Thus, 80% genes are not overlapping. Is there any trend that overlapping genes have higher score or non-overlapping genes show lower score in mRNA-seq analysis performed in this manuscript?

6. Line 256: regarding to the normalization, more details of the ChIP-seq protocol should be added in text and method sections. How was the normalization performed?

7. Line 256: ‘dRTEL1-GFP was found to be enriched at 654 genomic loci’ What the criteria to pick the genomic loci from the MACS2 output? The MACS2 output file would be helpful for readers if added in the supplementary. In addition, it should be mentioned in the main text that dRTEL1-GFP expressed not only in GSCs, but in both germline cells and somatic cells.

8. H2Av should be examined in ATM_tefu and CHK1_grp mutants as shown in CHK2_loki. 　In addition, p53 should be examined in CHK2_loki and the double KD/KO with dRTEL1. The epistatic hierarchy of DNA damage and DDR pathway genes can be more refined, by examining other components (ATM_tefu, CHK1_grp) of DDR pathway as well.

9. Involvement of the other DDR components (ATM_tefu, CHK1_grp) in the STAT92E mediated GSC maintenance could be addressed. If not, the authors should provide any argument for the abovementioned issue.

10. I wonder if overexpression of dETEL1 target genes would recover the defect of dETEL1 that gives STAT92E reduction and GSCs loss. Did the authors perform any experiments addressing this question? Alternatively, any argument could be provided for this concern.

11. L405-409; germline knockdown of most of the target genes except for Nipped-A in female exhibited no GSCs reduction. Knockdown efficiency should be addressed and shown.

12. Among the 654 target genes, 22 down-regulated genes and 49 up-regulated genes were identified. The criteria for the target genes should be described in the manuscript. In addition, I wonder whether the fold enrichment of ChIP binding has any relevance to the transcriptional level in RNA-seq data?

13. Line 616: ‘P-values and data significance was calculated according to two-tailed Student’s t-test.’ Where are these values shown? Although P-value is shown only for Nipped-A in text line 269, significance is not shown in Fig 5B.

14. Line 715: RNA-seq samples were prepared from ALH 60-72hr testes, while DNA samples for ChIP-seq were prepared from the adult testes. What is the reason why the different developmental stages were used? The reason can be mentioned in the main text.

15. Fig5C: the panel shows only one input. There may be another input sample to show.

Minor points:

1. Gal4 drivers should not be omitted in labels of the Figures; they should be written as tj-Gal4, nos-Gal4, and so on.

2. Fig2DE; With, Vasa signal seems to be reduced in Tj-Gal4> dRTEL1dsRNA, but not in bam-Gal4 > dRTEL1dsRNA. I wonder if somatic dRTEL1 knockdown have some effect on the germ cell viability or gene expression. Fig2H; ‘dRTEL1 shRNA’ should be ‘dsRNA’?

3. FigS3C: Describe the number of replicates and the standard deviations. Fig S3C legend: “same difference” > “some variations” ?

4. Line 257: ‘14.5% of the enrichment located within a 2-kb region upstream or downstream of gene coding region and 72.5% of the enrichment detected in the gene body region (Fig. 5A)’ How was this analysis performed? Describe the process in the text or method.

5. Fig4; label the specific gene names on it.

6. 9 down-regulated genes were identified by ChIP are essential in GSCs maintenance. Though qPCR shows their expression in testes, I wonder their expression patterns in testes—are they stage or cell-type specific? Have the authors examined those?

7. Check the Fig numbers are correct. L301: S5F and S5G>S5F-H?

8. L386-388; ‘The Drosophila ovarian GSCs are in direct contact with cap cells, contain an anteriorly-positioned spectrosome and express pMad.’ The ovarian GSC marker, pMad, should be properly introduced.

9. Fig7Q was not stated in the legend.

10. L407 does not correspond to the figures and genes. Fig8K is for Nipped-A.

11. Describe how the STAT92E intensity was quantified in Fig8L-O to induce Fig 8P.

12. S6L; p53 signal is faint. A better image should be provided.

13. Fig5A; percentiles of each category should be provided in the figure.

14. Fig5C: Add the Y-axis label. Is this a log scale? What does the arrow mean at the top of this panel? Is it a peak summit? If so, show the peak region, too.

15. Line 704: ‘using RESM software42’ Is this a correct description?

16. Line 717: ‘Two replicates were generated for control or experiment group.’ What is the control sample here? Is it the input DNA that is subjected to IP?

17. Line 721: ‘The chromatin was immune-precipitated by a ChIP-grade GFP antibody (ab290, Abcam). Sonication was conducted on Bioruptor sonicator’ Sonication must have been done before IP.

18. Fig4B: Add scale value of log2 fold change.

19. Table S3 lists 140 genes as “Genes identified to be required for germline development in Sanchez et al., 2016.” However, the text (Line 237) and Figure S3D describe that 136 genes are common between this study and previous study (Sanchez et al., 2016). Why are those numbers different? Similarly, 41 genes were listed in Table S3 but 38 genes were stated in text and Figure S3D.

**Reviewer #3**: (note: The reviewer accidentally pasted these into the Comments to Editor box instead of the Comments to Reviewer box so you may not see them online)

In this work, Yang and colleagues describe the role of dRTEL1 in the germline stem cells (GSCs) of the Drosophila testis (and ovary). The loss of GSCs in dRTEL1 mutants is clear and convincing, although clonal analysis should be done to better analyze the mutant phenotype, in particular their claim that loss of dRTEL1 induces premature GSC differentiation. In other experiments, they demonstrate that dRTEL1, which is involved in DNA repair and telomere maintenance in other systems, does indeed function through this pathway in the germline. My main concern is that dRTEL1 appears to be a general, conserved regulator of DNA repair and is likely doing this in most or all tissues of the fly. Thus, this is not a specific regulatory mechanism found in GSCs. It is not so surprising that blocking an essential cellular function like DNA repair would be detrimental for the cells in which it is blocked. It is a bit like saying that other factors essential for basic cellular processes, such as RNA polymerase or a cytoskeletal protein, are important in a particular cell type. It is nice to demonstrate this, but no one really doubted that such basic cell biology is important in most cells, and we don’t really learn anything new about the biology of that specific cell type. I feel that, once certain experiments like the clonal analysis are done, a subset of this work would be suitable for a more specialized journal than PLOS Genetics.

Other experiments in the paper are more problematic in my opinon. They conduct RNA-seq comparing wt to dRTEL mutant larval testes. However, they clearly document that the germline in dRTEL mutant testes is strongly reduced. Thus, this experiment is more likely to identify genes that are specific for the germline rather than genes altered more specifically by loss of dRTEL itself. They then go on to conduct CHIP-seq analysis with a tagged form of dRTEL. First of all, there is no evidence that this DNA repair protein can act as a transcription factor. Thus, CHIP-seq seems misguided at this stage. In addition, they claim a significant overlap between the RNA-seq and CHIP-seq analysis. However, while the RNA-seq analysis identified 14.9% of all genes as different between wt and dRTEL mutants, only 10.8 % of their CHIP-seq hits were also identified in the RNA-seq. Thus, there is actually a selection AGAINST the CHIP-seq targets in the RNA-seq data. Regardless of all of this, there is no evidence that suggests that dRTEL does anything other than function as a general factor in DNA repair. While the molecular mechanisms of DNA repair are certainly important to study, the fact that DNA repair is an important process for the germline, as it is for other cell types, is not significant enough on its own for publication in PLOS Genetics.

**Have all data underlying the figures and results presented in the manuscript been provided?**

Reviewer #1: Yes

Reviewer #2: Yes

Reviewer #3: Yes

PLOS authors have the option to publish the peer review history of their article (what does this mean?). If published, this will include your full peer review and any attached files.

Reviewer #1: No

Reviewer #2: No

Reviewer #3: No

---

## [Decision Letter · Decision Letter 1]

13 Sep 2021

Dear Dr Cai,

Thank you very much for submitting your Research Article entitled 'dRTEL1 Is Essential for the Maintenance of Drosophila male Germline Stem Cells' to PLOS Genetics.

The manuscript was fully evaluated at the editorial level and by independent peer reviewers. The reviewers appreciated the attention to an important topic but identified some concerns that we ask you address in a revised manuscript.

We therefore ask you to modify the manuscript according to the review recommendations. Your revisions should address the specific points made by each reviewer. Reviewer 1 had no requests and Reviewer 3 was unable to review this version. Reviewer 2 (comments attached) had several additional requests. I don't believe any of these really require additional experiments (points 1 and 3 may seem to, but only if you already have those experimental results available).

1) Provide a detailed list of your responses to the review comments and a description of the changes you have made in the manuscript. Please respond to all five points raised by Reviewer 2. I do not expect to send the manuscript out for review again.

[LINK]

Yours sincerely,

Jeff Sekelsky, PhD

Guest Editor

PLOS Genetics

Gregory P. Copenhaver

Editor-in-Chief

PLOS Genetics

Reviewer's Responses to Questions

**Comments to the Authors:**

Reviewer #1: Dear Authors,

Yang et al manuscript titled ‘dRTEL1 Is Essential for the Maintenance of Drosophila male Germline Stem Cells’ is a very compelling and comprehensive story indicating a novel function of Drosophila RTEL1 as a regulator of GSCs. This article, once

published, will be of great interest to the PLOS Genetics audience.

The authors utilized several different techniques to address their research questions. In the paper we can see the mastery of confocal imaging and traditional genetic approaches like gene knockdown and rescue, complemented by big data analyses

(ChIP-seq and RNA-seq). RTEL1 helicase, traditionally implicated in telomere maintenance and DNA repair, as demonstrated here, does appear to play a critical role in germline stem cell maintenance. This novel role for dRTEL1, although surprising, is supported by strong evidence. The authors thoroughly investigate RTEL1’s role in Drosophila testis GSCs, however, a brief look at the female germline indicates some elements of this regulation are conserved which present an opportunity for a follow-up publication.

I did not find any major issues preventing the publication of this manuscript. I appreciate authors’ attention to detail, figure labeling that focuses readers on the important elements of drosophila structures, and thorough controls, especially the use of cell linage-specific drivers.

Reviewer #2: uploaded.

**Have all data underlying the figures and results presented in the manuscript been provided?**

Reviewer #1: Yes

Reviewer #2: Yes

PLOS authors have the option to publish the peer review history of their article (what does this mean?). If published, this will include your full peer review and any attached files.

Reviewer #1: No

Reviewer #2: No

---

## [Editor Report · Decision Letter 2]

23 Sep 2021

Dear Dr Cai,

We are pleased to inform you that your manuscript entitled "dRTEL1 Is Essential for the Maintenance of Drosophila male Germline Stem Cells" has been editorially accepted for publication in PLOS Genetics. Congratulations!

Yours sincerely,

Jeff Sekelsky, PhD

Guest Editor

PLOS Genetics

Gregory P. Copenhaver

Editor-in-Chief

PLOS Genetics

**Data Deposition**

http://datadryad.org/submit?journalID=pgenetics&manu=PGENETICS-D-21-00484R2

**Press Queries**

---

## [Editor Report · Acceptance letter]

29 Sep 2021

PGENETICS-D-21-00484R2 

dRTEL1 Is Essential for the Maintenance of Drosophila male Germline Stem Cells 

Dear Dr Cai, 

We are pleased to inform you that your manuscript entitled "dRTEL1 Is Essential for the Maintenance of Drosophila male Germline Stem Cells" has been formally accepted for publication in PLOS Genetics! Your manuscript is now with our production department and you will be notified of the publication date in due course.

With kind regards,

Zsofia Freund

PLOS Genetics

On behalf of:
